# Aberrant DNA methylation distorts developmental trajectories in atypical teratoid/rhabdoid tumors

Meeri Pekkarinen[1,*], Kristiina Nordfors[2,3,10,*], Joonas Uusi-Mäkelä[1,*], Ville Kytölä[1,*], Anja Hartewig[1], Laura Huhtala[1], Minna Rauhala[3,4], Henna Urhonen[1], Sergei Häyrynen[1], Ebrahim Afyounian[1], Olli Yli-Harja[5,6], Wei Zhang[7], Pauli Helen[4], Olli Lohi[2,3,5], Hannu Haapasalo[5,8], Joonas Haapasalo[3,4,8], Matti Nykter[1], Juha Kesseli[1], Kirsi J Rautajoki[1,9]

Atypical teratoid/rhabdoid tumors (AT/RTs) are pediatric brain tumors known for their aggressiveness and aberrant but still unresolved epigenetic regulation. To better understand their malignancy, we investigated how AT/RT-specific DNA hypermethylation was associated with gene expression and altered transcription factor binding and how it is linked to upstream regulation. Medulloblastomas, choroid plexus tumors, pluripotent stem cells, and fetal brain were used as references. A part of the genomic regions, which were hypermethylated in AT/RTs similarly as in pluripotent stem cells and demethylated in the fetal brain, were targeted by neural transcriptional regulators. AT/RT-unique DNA hypermethylation was associated with polycomb repressive complex 2 and linked to suppressed genes with a role in neural development and tumorigenesis. Activity of the several NEUROG/NEUROD pioneer factors, which are unable to bind to methylated DNA, was compromised via the suppressed expression or DNA hypermethylation of their target sites, which was also experimentally validated for NEUROD1 in medulloblastomas and AT/RT samples. These results highlight and characterize the role of DNA hypermethylation in AT/RT malignancy and halted neural cell differentiation.

## Introduction

Central nervous system (CNS) tumors can arise at any age and have the highest cancer-associated mortality rate in pediatric patients (Ostrom et al, 2019). Atypical teratoid/rhabdoid tumors (AT/RTs), medulloblastomas (MBs), and choroid plexus tumors (PLEXs) are CNS tumors detected in infants (Burger et al, 1998), though they can also occur in adults. AT/RTs and MBs are aggressive, grade 4 embryonal tumors according to the World Health Organization (WHO) classification (Louis et al, 2021), and most aggressive (grade III) PLEXs, namely, choroid plexus carcinomas, are malignant and associated with poor overall survival rates (Wolff et al, 2002; Louis et al, 2021). Improving patient outcomes for these aggressive tumors is an urgent task.

Aberrant epigenetic regulation is often driving malignancy in pediatric tumors (Faria et al, 2011; Erkek et al, 2019). For AT/RTs, the significance of epigenetics is highlighted by the sole recurrent genetic alteration in their genome, namely, the inactivation of *SMARCB1* or *SMARCA4*. Both are subunits of the mammalian SWItch/Sucrose Non-Fermentable (SWI/SNF) chromatin remodeling complex. SWI/SNF is critical for the targeted opening of chromatin during neural development (Sokpor et al, 2017) via EP300-mediated histone 3 lysine 27 acetylation (H3K27ac) (Sokpor et al, 2017). Polycomb repressive complex 2 (PRC2) is considered an antagonist for the SWI/SNF complex, as its key subunit EZH2 trimethylates H3K27 (leading to H3K27me3) and silences chromatin. However, H3K27me3 is also depleted in AT/RTs (Erkek et al, 2019), suggesting an H3K27me3-independent epigenetic driver for AT/RT development.

DNA methylation and other epigenetic regulation have been studied with an increasing interest in cancer. DNA methylation contributes to cell differentiation (Stricker & Götz, 2018); is often aberrant in malignancies (Saghafinia et al, 2018); and is, together with histone methylation, a clinically relevant therapeutic target (Sato et al, 2017; Patnaik & Anupriya, 2019). Furthermore, it is used for the accurate classification of CNS tumors and early cancer detection (Capper et al, 2018; van der Pol & Mouliere, 2019). The classifier also reveals tumor type– or subtype-specific DNA

[1]Prostate Cancer Research Center, Faculty of Medicine and Health Technology, Tampere University and Tays Cancer Center, Tampere University Hospital, Tampere, Finland [2]Tampere Center for Child Health Research, Tays Cancer Center, Tampere University and Tampere University Hospital, Tampere, Finland [3]Tays Cancer Center, Tampere University Hospital, Tampere, Finland [4]Department of Neurosurgery, Tays Cancer Centre, Tampere University Hospital and Tampere University, Tampere, Finland [5]Faculty of Medicine and Health Technology, Tampere University and Tays Cancer Center, Tampere University Hospital, Tampere, Finland [6]Institute for Systems Biology, Seattle, WA, USA [7]Cancer Genomics and Precision Oncology, Wake Forest Baptist Comprehensive Cancer Center, Winston-Salem, NC, USA [8]Fimlab Laboratories Ltd, Tampere University Hospital, Tampere, Finland [9]Tampere Institute for Advanced Study, Tampere University, Tampere, Finland [10]Unit of Pediatric Hematology and Oncology, Tampere University Hospital, Tampere, Finland

Correspondence: kirsi.rautajoki@tuni.fi
*Meeri Pekkarinen, Kristiina Nordfors, Joonas Uusi-Mäkelä, and Ville Kytölä contributed equally to this work

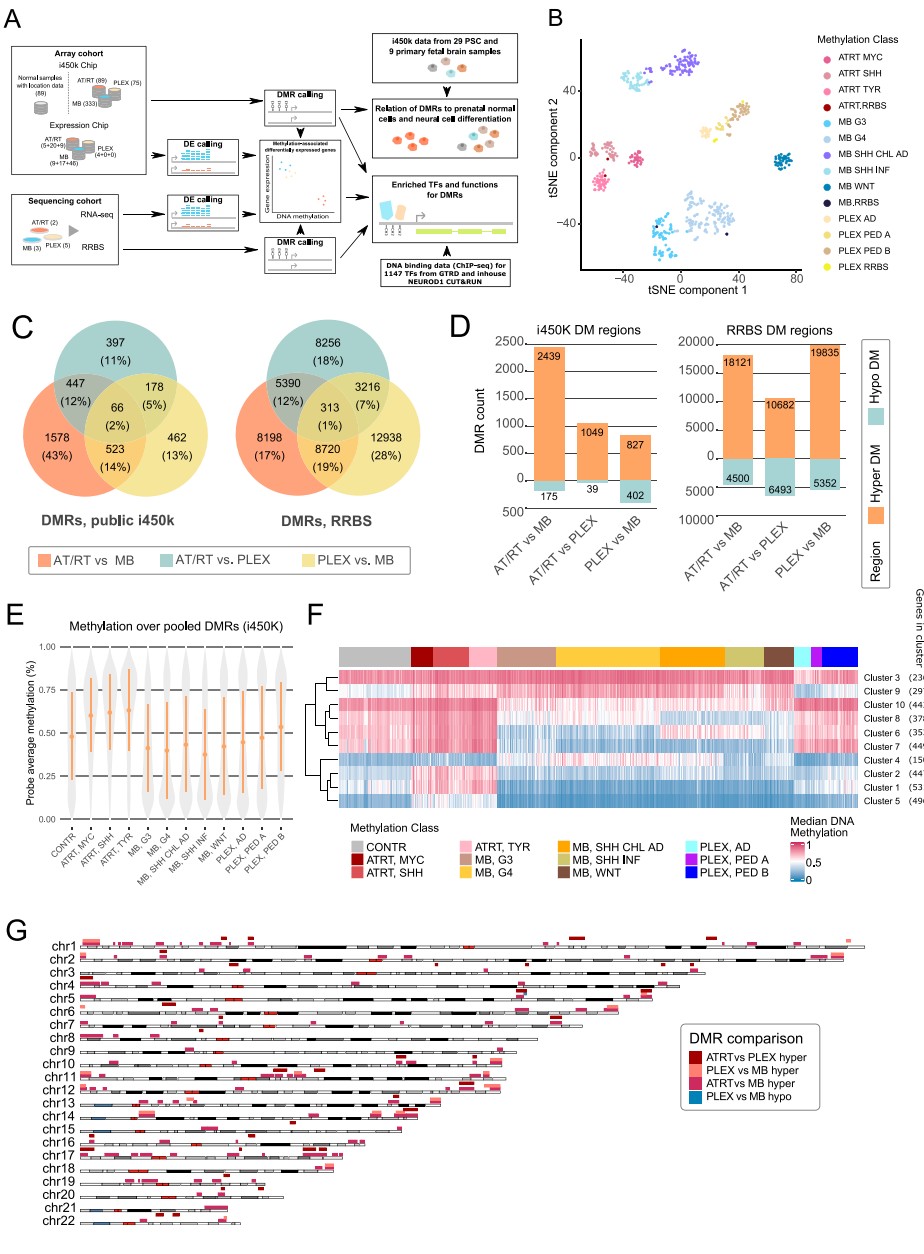

**Figure 1. Characterization of DNA methylation differences among AT/RTs, MBs, and PLEXs reveals AT/RT hypermethylation across all AT/RT subclasses and the genomic regions affected by large-scale DNA methylation changes.**
**(A)** Illustration of our data analysis and integration approach. The number of samples in each cohort is shown on the left. Data were used to call differentially methylated regions (DMRs) and differentially expressed (DE) genes. Data from the Gene Transcription Regulation Database provided transcription factor DNA binding information. DNA methylation data from pluripotent stem cells and normal samples were used as references and to study normal neural cell differentiation. **(B)** Tumor types are separated into tumor subgroups (omitted from Capper et al [2018]) based on DNA methylation, when the 10,000 most variable regions measured in both i450k and RRBS data (see the Materials and Methods section) were used for the tSNE visualization. RRBS samples are positioned adjacent to i450k samples representing the tumor subgroups that matched their clinical diagnosis. **(C)** Venn diagrams showing the number of DMRs in each comparison. Tumor type–specific DMRs are marked into the intersecting areas. A higher number of DMRs were detected in the RRBS than in the i450k data. For i450k results, DMRs were filtered using DNA methylation data from normal brain samples. **(D)** For AT/RTs, larger numbers of hypermethylated than hypomethylated regions were detected in all the comparisons in both i450k and RRBS data. The numbers of DMRs and direction of DNA methylation change for each comparison in both datasets. **(E)** AT/RT subgroups showed the highest DNA methylation among the pooled DMRs when compared to other tumor types and normal control samples (CONTR). Average DNA methylation of the probes hitting each i450k DMR is visualized as tumor subgroup-wise violin plots. **(F)** k-Means clustering analysis revealed DMR clusters that are specifically hypermethylated in AT/RTs. The median DNA methylation of the DMRs in each cluster was used to summarize the DNA methylation patterns. None of the DMR clusters showed AT/RT subtype–specific DNA methylation patterns, but there were DNA methylation differences between tumor subtypes within both MBs and PLEXs. **(G)** Several topologically associating domains were influenced by large-scale DNA methylation differences, especially in the AT/RT-MB comparison. Karyoplot visualizes the topologically associating domains that harbor large-scale DNA methylation differences, that is, several DMRs that were predominantly either hyper- or hypomethylated in the comparison (see the Materials and Methods section). Color indicates a comparison in which a difference was observed.

methylation patterns, linking DNA methylation to tumor type–specific oncogenesis (Capper et al, 2018).

Anomalous cell differentiation is typical for cancer, because less differentiated cell features can be hijacked by tumor cells (Hanahan, 2022). A low differentiation state provides an evolutionary advantage and drives several hallmarks of cancer (Hanahan, 2022). A better understanding of the mechanisms that allow malignant cells to resist cell differentiation, which contributes to their aggressiveness, is needed (Atlasi & Stunnenberg, 2017).

Traditionally, researchers have compared and analyzed subtypes of AT/RTs, MBs, and PLEXs separately (Merino et al, 2015;

Torchia et al, 2015; Johann et al, 2016; Northcott et al, 2017). Based on the DNA methylation data, different AT/RT and PLEX methylation subtypes are located next to each other in t-distributed stochastic neighbor embedding (tSNE) visualization, suggesting lower heterogeneity within the tumor type, whereas MBs are more heterogeneous: they represent very distinct DNA methylation patterns and clear separation of subtypes (Capper et al, 2018). In this study, we set out to reveal and functionally evaluate the hallmark DNA methylation in AT/RTs. MBs and PLEXs have a partly shared history and characteristics with AT/RTs (Biegel et al, 1999; Gessi et al, 2003; Judkins et al, 2005), and they represent different developmental

lineages (Faria et al, 2011; Louis et al, 2021), which provides the good setup for comparison. They were used, together with pluripotent stem cells (PSCs) and fetal brain (FB) samples, as external references to facilitate AT/RT-specific feature detection.

# Results

### DNA is hypermethylated in AT/RTs when compared to MBs or PLEXs

To study oncogenic epigenetic regulation in AT/RTs, we collected genome-wide DNA methylation Illumina microarray data (i450K) from 497 tumors and unmatched microarray expression data from 110 tumors; 89 normal brain DNA methylation samples were used as controls in i450k-based DNA methylation analysis. In addition, we generated matched data from 10 tumors with reduced representation bisulfite sequencing (RRBS) and RNA sequencing (RNA-seq) (Fig 1A, Table S1). We thus conducted two distinct analyses, one based on array data and the other on sequencing data, and used them to select the method-independent results in various subsequent analyses. tSNE analysis identified and confirmed the tumor types and subtypes of the cohorts (Fig 1B). MB samples were separated into WNT, SHH child/adult, SHH infant, group 3, and group 4 entities (Capper et al, 2018), largely following the latest 2021 WHO classification (Louis et al, 2021), whereas AT/RT and PLEX subtypes were closely positioned, as expected. Our in-house RRBS samples were positioned near the microarray sample clusters representing their clinical diagnosis.

Differential DNA methylation analyses between AT/RTs, MBs, and PLEXs resulted in 4,931 and 64,983 differentially methylated regions (DMRs) in the i450k (methylation difference: beta fold change >= 0.20; see details from the Materials and Methods section) and RRBS (methylation difference: beta fold change >= 0.25) data, respectively (Fig 1C). Higher DMR counts in RRBS data analysis were predictable because i450k DMRs were, on average, longer and RRBS has a wider genomic coverage (Meissner et al, 2005; Pidsley et al, 2013). Tumor location information and normal samples were used to reduce tumor location–related differences in i450k DNA methylation data (see the Materials and Methods section), decreasing the DMR counts, as well (Fig 1C versus Fig S1A). There were a higher proportion of intergenic DMRs in RRBS than in i450k data (Fig S1B, Table S1). DMRs were distributed across all chromosomes (Fig S1C). Most (>70%) of the DMRs were more methylated in AT/RTs and less methylated in MBs than in the other tumor types (Fig 1D). Consistently, the highest DNA methylation levels were detected in AT/RTs and lowest in MBs (Figs 1E and S1D and E).

In i450k data with a higher sample count, DNA methylation in DMRs was consistently higher in the AT/RT subgroups (MYC, SHH, and TYR) than in the subgroups of other tumor types (Wilcoxon rank sum test, $P < 0.001$, effect size Cliff's delta $|\delta| > 0.2$ for all comparisons except 0.18 for AT/RT-SHH versus PLEX, PED B and 0.13 for AT/RT-MYC versus PLEX, PED B) (Fig 1E, Table S1). In PLEXs, the highest DNA methylation was detected in the most aggressive subgroup, PLEX, PED B (Fig 1E). To subcategorize the i450k DMRs, we ran k-means clustering, which resulted in 10 clusters (Figs 1F and

S2), of which three were ATRT-specific (clusters 1, 2, and 5), one PLEX-specific (cluster 9), and two MB-specific (clusters 7 and 10). Clustering also revealed tumor subgroup–specific DNA methylation patterns, especially for MBs (clusters 4, 6, and 8) and the AD subgroup of PLEXs. However, ATRT subgroups behaved similarly across clusters (Fig 1F) and also predominantly across individual DNA methylation sites (Fig S2A–J).

DNA methylation can be focal or spread to larger genomic regions, which are typically bordered by topologically associating domain (TAD) boundaries (Szabo et al, 2019). These two methylation types have different regulatory outcomes (Buitrago et al, 2021). The DMRs present in both RRBS and i450k data were involved in large-scale DNA methylation differences within specific TADs, especially in chromosomes 1, 11, 13, 17, and 19 in the AT/RT-MB comparison (Fig 1G).

### Distinct neural differentiation–related transcriptional regulators are enriched in AT/RT- and MB-hypermethylated sites

To study the function and possible upstream regulation of regions showing differential DNA methylation, we used the chromatin immunoprecipitation sequencing (ChIP-seq) data collected from the Gene Transcription Regulation Database (GTRD) (Yevshin et al, 2019) for identifying the transcription factors and other transcriptional regulators (referred to as TFs from now on) whose binding sites are over-represented in the analyzed DMRs (adjusted $P < 0.05$, one-sided Fisher's exact test). To focus our analysis on cancer-specific epigenetic regulation, we included only DMRs, which were hypermethylated or hypomethylated in a cancer-specific manner; that is, they behaved similarly in both comparisons (being, e.g., hypermethylated in AT/RTs when separately compared to MBs and to PLEXs) (Fig 2A). Only TFs enriched with both i450k and RRBS-based analyses were analyzed further.

Of the 183 enriched TFs, 136 (74%) were unique to one tumor type, 30 (16%) associated with two tumor types, and 17 (9.3%) to all three (Fig 2B, Table S2). Altogether, 77 TFs were enriched in AT/RT-specific hypermethylated (AT/RT-hyper) DMRs, 114 TFs in MB-hyper DMRs, and 10 TFs in MB-hypomethylated (MB-hypo) DMRs (Fig 2B). A higher proportion of MB-hyper DMRs were located in CpG islands (Fig S3A, Table S1), which might partly explain a high number of TFs enriched in these regions. No AT/RT subgroup specificity was observed in the DNA methylation patterns of differentially methylated TF-binding sites (TFBSs) (selected examples in Fig S3B). DNA methylation has been shown to reduce the DNA binding of 33/49 (67%) TFs with DNA binding activity and curated information (Table S2) (Yin et al, 2017).

To summarize the regulatory roles of the enriched TFs, we used a literature search to functionally annotate the TFs into different themes (Table S2), which were chosen based on TF functions and gene families that recurrently popped up when manually going through the enriched TFs in the first phase of the analysis (Merino et al, 2015; Swartling et al, 2015; Northcott et al, 2017; Erkek et al, 2019). Notably, 15 neural differentiation and neural cell–related TFs were uniquely enriched in AT/RT-hyper DMRs (Fig 2B). These included NEUROD1, ASCL1, and MYCN. When reanalyzing the TF binding after dividing the DMRs into clusters with k-means clustering (Fig 1F), AT/RT-unique TFs BCHE, MYCN, NFIB, NFIC, and PITX3 were enriched in AT/RT-specific DMR clusters 1 and/or 5 but none in

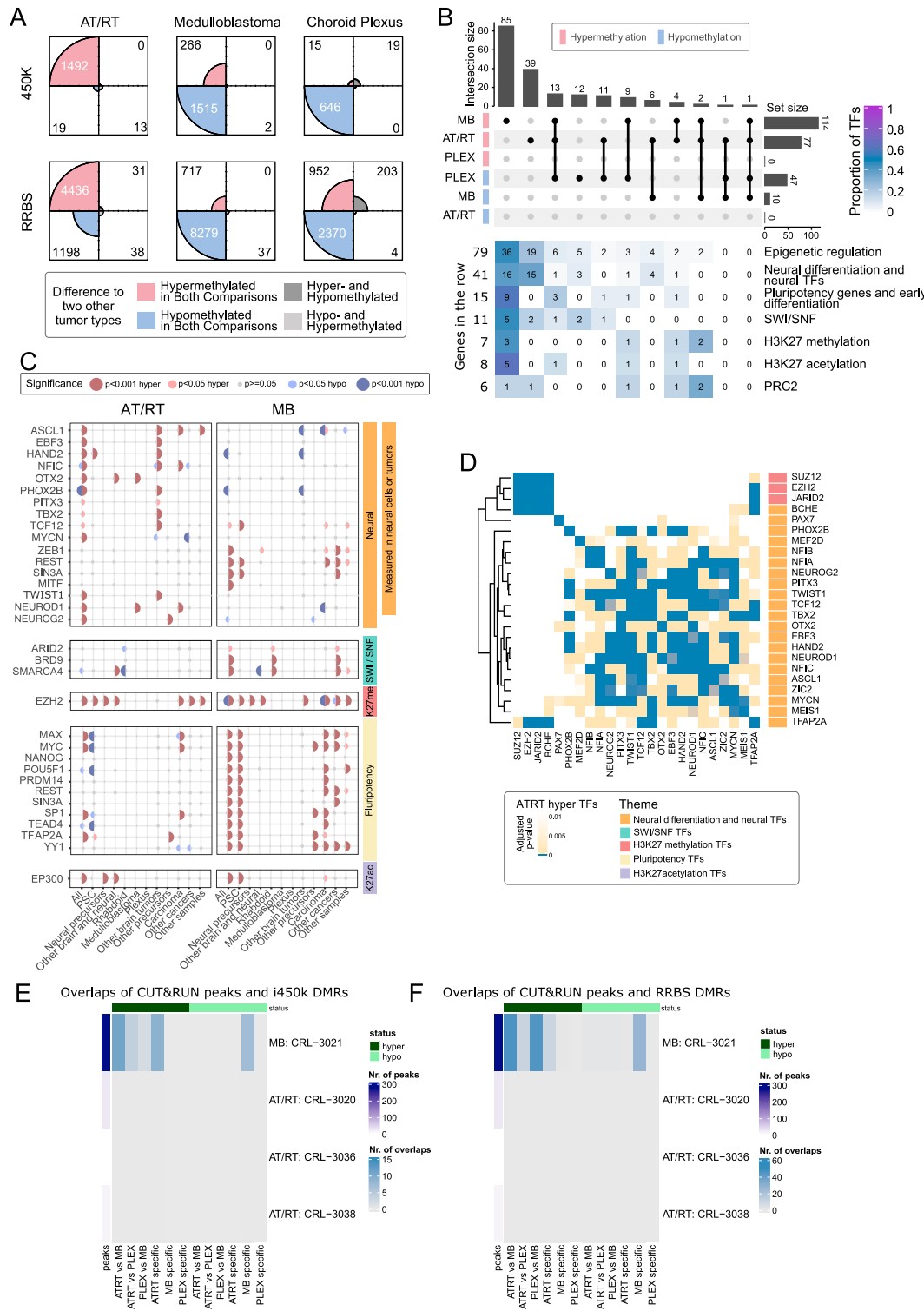

**Figure 2. Neural cell differentiation–related TFs and key epigenetic regulators are enriched in tumor type–specific DMRs.**
**(A)** Most of the AT/RT-specific DMRs were hypermethylated (99% and 79% in i450K and RRBS data, respectively), whereas hypomethylated DMRs were more commonly observed in MBs (85% and 88% in i450K and RRBS data, respectively) and PLEXs (98% and 71% in i450K and RRBS data, respectively). Four-field plots for each tumor type show the number of hypermethylated and hypomethylated regions with respect to two other tumor types. Tumor-specific DMRs have the DNA methylation change in the same direction when compared to two other tumor types (e.g., hypermethylated in AT/RTs when compared to MBs and to PLEXs). **(B)** Most of the transcription factors and other transcriptional regulators (jointly referred to as TFs) are specifically enriched in AT/RT-hyper, MB-hyper, or PLEX-hypo DMRs and largely linked to neural differentiation, SWI/SNF, and PRC2. Upper part: upset plot showing the number of enriched TFs for regions that are hypermethylated or hypomethylated in an AT/RT-, MB-, and PLEX-specific manner. Some transcriptional regulators were enriched in several tumor-specific DMR groups. Lower part: the number of TFs in manually annotated

AT/RT-specific cluster 2 (Table S2). Four neural differentiation–related TFs were enriched in both AT/RT-hyper and MB-hypo DMRs (Fig 2B, Table S2). Of these, NEUROG2 and HAND2 were enriched in cluster 2, whereas PHOX2B and PAX7 were enriched in clusters 1 and/or 5 (Fig 1F, Table S2).

We also noticed that 16 neural differentiation–related TFs were uniquely enriched in MB-hyper sites (Fig 2B). However, five of these (REST, SIN3A, ZEB1, TBX1, and ZHX2) inhibit neural differentiation (Table S2). In addition, four of the remaining TFs were active in the later phases of neural differentiation, whereas 7 of 15 neural TFs enriched in ATRT-hyper sites were active in earlier stages (during differentiation from PSCs and from neural stem cells) (Table S2). Thus, neural cell differentiation–related TFs in AT/RTs and MBs represent different functions.

When considering both RNA-seq and array expression data, the expression of 164 enriched TFs (90%) was not significantly associated with any tumor type (Figs S4A and B and S5A and B, Table S3). ERF was the only differentially expressed (DE) TF in AT/RTs compared with other tumors (adjusted $P < 0.01$, absolute logFC > 1), whereas 13 TFs, including EBF3 and NEUROD1, showed MB-specific differential expression. Neural differentiation TFs enriched in AT/RT-hypermethylated regions were expressed in AT/RTs, and some TFs (e.g., *NEUROG2* and *HAND2*) even showed the highest expression in AT/RTs (Fig S5A).

GTRD data originate from a variety of cell types, and TF binding is also cell type–dependent (Yevshin et al, 2019). To take into account the stage of neural development, brain tumor–derived data, and cancer association, we categorized GTRD data based on the sample type in which the binding data of the TFs have been obtained. This resulted in 11 sample groups, which represented our analyzed tumor types, other brain tumors, neural progenitors, and more differentiated neural sample types, as well as PSCs, other tumor types, and all the other sample types (Table S2) (Yevshin et al, 2019). We then reran the TFBS enrichment analysis for cancer-specific DMRs using these 11 binding site subsets (called as categorized GTRD analysis from now on) and summarized the obtained results as a dot plot (Fig 2C, full version in Fig S6A and B). This approach helped us to investigate whether DNA hypermethylation was mainly covering TFBSs in neural cells, so sites that are bound by the TF in (more) differentiated cells of the neural lineage, whether TF binding has been measured from the analyzed tumor types, and whether TFs are shown to bind to these sites already in PSCs that are expected to represent the earliest phases of (neural) cell differentiation. Altogether, 16 of the analyzed TFs were both neural differentiation– and neural cell–related and measured in brain tumors or neural cells (Fig 2C). Of these 16 TFs, 11 (69%), including

NEUROD1, were enriched in sites specifically hypermethylated in AT/RTs (AT/RT-hyper sites), MYCN was enriched in AT/RT-hypo sites, and ZEB1 was enriched in MB-hyper regions (Fig 2C, Table S2). On the contrary, 12 of 16 (75%) neural TFs enriched only in MB-hyper sites were measured or significant only in pluripotent cells or non-neural cell types (Fig S6A and B, Table S2). These included REST and SIN3A, which inhibit neural differentiation and whose binding sites in PSCs, but not those in "other brain and neural samples" or "other brain tumors," were enriched in MB-hyper regions. Finally, the binding sites of 9 of 85 (11%) TFs uniquely enriched in MB-hyper regions were related to the PSC state and early differentiation (Fig 2B), and they significantly overlapped MB-hyper DMRs when TF binding was measured in PSCs (Fig 2C, Table S2). This enrichment of pluripotency factors in MB-hyper regions is implying that pluripotency programs are not active or are disrupted in MB cells, making it more likely that programs related to more mature cell states have taken over. Together, these findings suggest that the binding sites of neural TFs are hypermethylated in AT/RTs, which might interfere with their DNA binding, whereas TFBSs in PSCs are hypermethylated in MBs, thus representing a more advanced neural differentiation state.

## DNA binding sites of SWI/SNF, PRC2, and EP300 harbor unique DNA methylation patterns in AT/RTs

As the SWI/SNF chromatin remodeling complex is incomplete in AT/RTs because of the full inactivation of *SMARCB1* or *SMARCA4*, we separately analyzed its subunits in the categorized GTRD data. Interestingly, the binding sites of SMARCA4- and the PBAF (polybromo-associated BAF, mammalian SWI/SNF complex)-specific TF ARID2 in rhabdoid tumor samples were enriched in AT/RT-hypo and MB-hyper regions (Fig 2C) (Mashtalir et al, 2018), whereas SMARCA4 binding sites in other brain and neural cells were enriched in AT/RT-hyper and MB-hypo regions (Fig 2C, Table S2). Consistently, the acetyltransferase EP300 binding sites in normal neural precursor and differentiated neural cells were hypermethylated in AT/RTs, and EP300 binding sites in pluripotent cells were hypermethylated in MBs (Fig 2C). These results suggest that the neural SWI/SNF-EP300 target sites have been methylated in AT/RTs and unmethylated in MBs. Furthermore, the SWI/SNF complex remains able to promote active chromatin in the regions to which it binds even in the absence of *SMARCB1*.

PRC2 and EZH2 are antagonizing SWI/SNF and have been reported to populate neural SWI/SNF binding sites in AT/RTs (Erkek et al, 2019). In our analysis, EZH2 was enriched in AT/RT-hyper, MB-hyper, or MB-hypo regions in the uncategorized GTRD analysis,

---

function-related theme groups is shown for each upset plot column. The color of the heatmap shows the fraction of TFs in each theme (row). **(C)** Binding sites of neural TFs measured in brain tumors and other neural samples were enriched in AT/RT-hyper DMRs, whereas those measured in pluripotent stem cells were enriched in MB-hyper DMRs. GTRD TF binding data were categorized based on the measured sample type into the listed subsets (at the bottom of the plot), and the enrichment of TF binding sites in all the DMRs with AT/RT- or MB-specific DNA methylation was calculated for each of the GTRD subsets separately. Category "All*" means all the reported TF binding sites, so the full GTRD data. Results for the most relevant TFs are shown after organizing them into the theme groups listed in Fig 2B. The dot is not marked when a given TF or other regulator is not measured in a given GTRD subset. **(D)** PRC2 subunits rarely co-localize with neural TFs and other regulators in AT/RT-hypermethylated sites. Heatmap visualization of the enrichment *P*-value (one-sided Fisher's exact test) for co-localization. All the adjusted *P*-values of 0.01 or higher are marked in white. Themes for each TF are annotated on the right-hand side of the heatmap. **(E, F)** In our CUT&RUN sequencing analysis, the DNA binding sites of NEUROD1 in the MB cell line overlapped regions that are hypermethylated in AT/RTs and hypomethylated in MBs (AT/RT versus MB-hyper), whereas no NEUROD1 binding sites in AT/RT samples overlapped with DMRs. Heatmaps showing the NEUROD1 binding sites located in different types of DMRs in i450K (E) and RRBS (F) data across the analyzed cell lines.

suggesting different modes of regulation (Fig 2C, Table S2). When we analyzed TF binding in AT/RT-hyper regions after dividing them into DMR clusters (Fig 1F), EZH2 was enriched in cluster 1, which shows the highest DNA methylation in AT/RTs and the lowest in MBs (Table S2). When examining the TF enrichment results in the categorized GTRD data, all the EZH2 binding sites, but those originating from other brain tumors, were enriched in AT/RT-hyper regions, including EZH2 binding sites in neural progenitors and more differentiated neural samples (Fig 2C).

### PRC2 subunits and neural differentiation–related TFs rarely co-localize

We performed TF binding co-localization analysis for ATRT-hyper, MB-hyper, MB-hypo, and PLEX-hypo DMRs (Figs 2D, S7A–C, and S8). Most TFs related to neural differentiation share their binding sites and were clustered, but significant co-localization (adjusted $P < 0.01$, one-sided Fisher's exact test) was rarely observed between PRC2 members and TFs related to neural differentiation (Figs 2D, S7A–C, and S8). In AT/RT-hyper regions, only the neural TFs BCHE and TFAP2A co-localized with PRC2 subunits. TFAP2A is known to regulate neural crest induction and cranial placode specification (Wang et al, 2011). BCHE is expressed early in the nervous system development and plays a role in neural stem cell development (Tiethof et al, 2018). The results suggest that PRC2-related DNA methylation largely affects parts of the genome other than the sites that recruit neural TFs during differentiation, BCHE and TFAP2A representing possible TFs that act at their intersection.

### CUT&RUN reveals reduced binding of NEUROD1 in AT/RT DNA–hypermethylated sites

To experimentally validate our hypothesis of inadequate DNA binding of NEUROD1 in AT/RTs, we performed cleavage under targets and release using nuclease (CUT&RUN), sequencing experiment targeting NEUROD1 in one MB and three AT/RT cell lines (see the Materials and Methods section). For the analysis, we selected only the CUT&RUN peaks that overlapped with any of the previously reported NEUROD1 binding sites in the GTRD, or contained NEUROD1 binding motif in the sequence (Grant et al, 2011; Yevshin et al, 2019), or fulfilled both of these criteria (Fig S9A). We detected less binding of NEUROD1 in all of the three AT/RT samples (28, 1, and 12 peaks) compared with the MB sample (275 peaks) (Table S2). None of the peaks detected in AT/RT samples were both reported in the GTRD and to having NEUROD1 binding motif, whereas 215 (78%) peaks detected in the MB sample fulfilled both criteria (Fig S9A). Nearly all (273, 99.3%) of NEUROD1 binding sites detected in MBs were detected only in that sample in our CUT&RUN analysis (Fig S9B). NEUROD1 binding sites detected in AT/RT samples harbored no overlap with any of the DMRs or regions with cancer-specific DNA methylation (Figs 2E and F and S9C and D, Table S2). However, in the MB sample, NEUROD1 was binding to DMRs that were differentially methylated in the AT/RT versus MB comparison or were AT/RT-specific or MB-specific. The vast majority of sites in the AT/RT versus MB comparison (46/48 and 11/11 in RRBS and i450k data, respectively) harbored DNA hypermethylation in AT/RTs (Figs 2E and F and S9C and D, Table S2). This supports our

previous claims, as we see NEUROD1 binding at sites hypomethylated in MBs. Similarly, the NEUROD1 binding sites in the MB sample that were overlapping with AT/RT-specific DMRs were predominantly hypermethylated in AT/RTs when compared to MBs and PLEXs (10/11 and 9/9 in RRBS and i450k data, respectively) and those overlapping with MB-specific DMRs were predominantly hypomethylated in MBs when compared to AT/RTs and PLEXs (27/28 and 7/7 in RRBS and i450k data, respectively), further supporting our hypothesis (Fig 2E and F, Table S2). Furthermore, we generated RRBS data from the cell lines used in the CUT&RUN analysis, which also confirmed that most of the NEUROD1 binding sites, which were detected in the MB sample, show high DNA methylation in all the AT/RT samples (beta value >0.5 in all the AT/RT samples in 258 of 275 [94%] sites) (Fig S9E). Although NEUROD1 RNA expression is on average higher in MBs than in ATRT (Fig S5), its protein levels were similar in all the studied cell lines (Fig S9F), so the differential expression of NEUROD1 cannot explain the observed differences in DNA binding.

Together, these findings support the notion that DNA methylation in AT/RTs is disrupting the binding of DNA methylation–sensitive TFs to these sites, thus altering or preventing the activation of neural differentiation–related programs.

### AT/RTs harbor both unique and PSC-like DNA methylation patterns

Next, we analyzed DMRs in the context of normal neural cell differentiation using i450k DNA methylation data from PSC and primary FB samples (Nazor et al, 2012; Colunga et al, 2019) (Fig 3A). This allowed us to define whether tumor-specific DNA methylation levels actually reflect either low (PSC-like) or higher (FB-like) cell differentiation state or are unique to each tumor type, and whether they are altered during the differentiation from PSCs to FB. When visualizing the DNA methylation levels in DMRs (Fig 1E), the AT/RT was most similar to PSCs (Fig 3B). We observed MB-unique, AT/RT-unique, PSC-like, and FB-like DMRs, part of which change their DNA methylation levels during normal neural differentiation (Fig 3C, left side, Fig S10A, Table S3). Of the 898 DMRs hypermethylated in AT/RTs, 322 (39%) were similarly methylated in PSCs as in AT/RTs and demethylated during differentiation from PSCs to FB (group 3, Fig 3C). Only 31 (3.5%) of AT/RT-hypermethylated DMRs were FB-like (group 5), highlighting the higher similarity of AT/RTs to PSCs than FB. However, there were also 300 (33%) regions with AT/RT-unique DNA hypermethylation. Of these, 124 (14%) were not related to differentiation (group 2) and 176 (20%) demethylated during neural differentiation (group 1). To summarize, 75% of AT/RT DMRs were demethylated upon neural differentiation, and they include both PSC-like (39%) and AT/RT-unique (61%) regions (Fig 3C).

In contrast, 94% MB-hyper DMRs and 73% MB-hypo DMRs were unique to MBs (Fig 3C, Table S3). Only 17 (15%) unique MB-hyper and as many as 445 (48%) unique MB-hypo DMRs were hypomethylated during differentiation (including 321 [34%] DMRs, which were also PSC-like in the AT/RT comparison). Furthermore, 221 (25%) MB-hypomethylated DMRs were FB-like and hypomethylated during differentiation (group 13). Thus, a high proportion of even tumor type–unique DMRs change their DNA methylation during differentiation

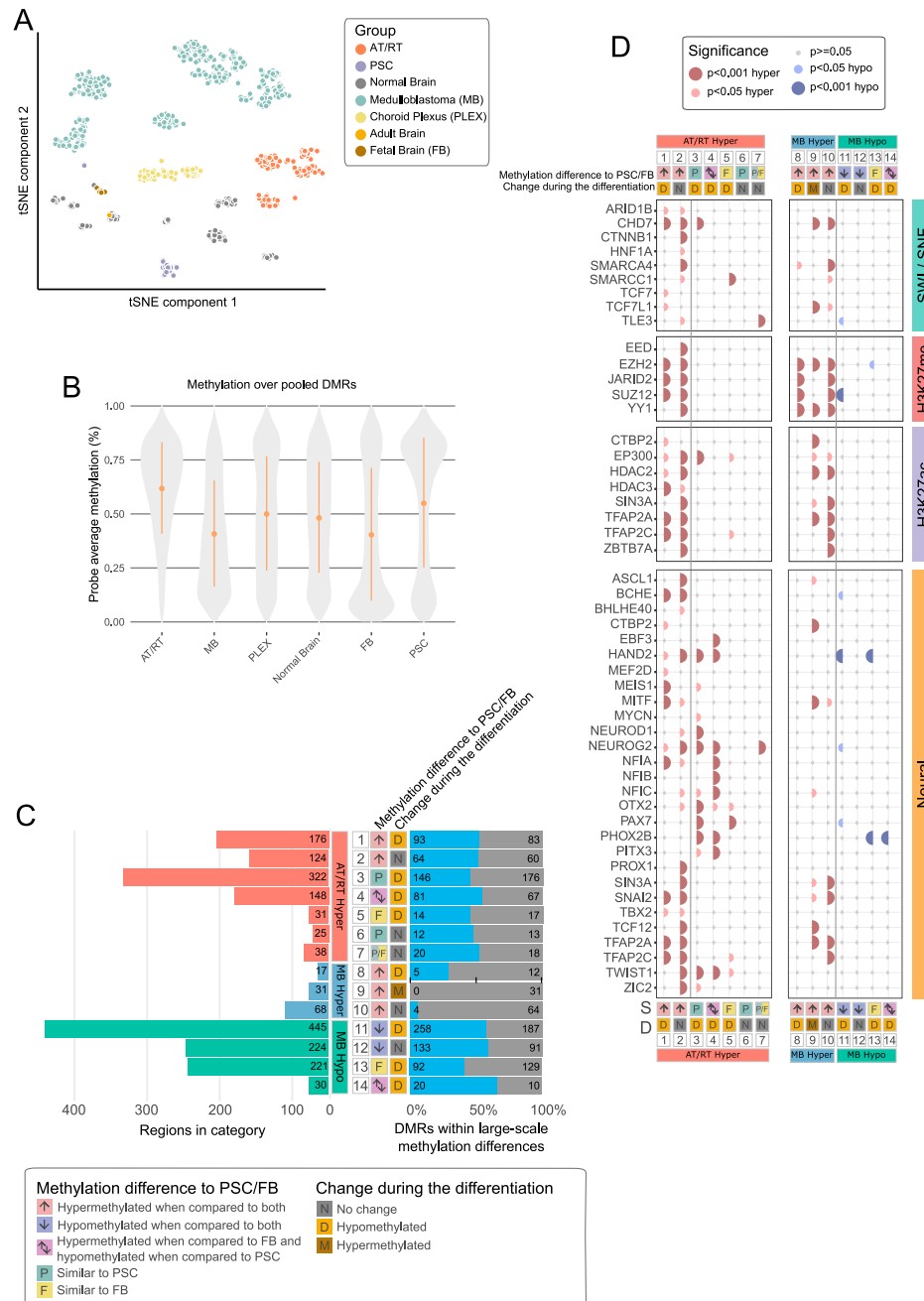

**Figure 3. AT/RTs harbor pluripotent stem cell–like and AT/RT-unique DMRs, which are associated with the DNA binding of relevant transcriptional regulators.**
**(A)** Pluripotent stem cells (PSCs), primary adult brain, and primary fetal brain (FB) are separated from tumor samples based on DNA methylation in tSNE visualization, when the 10,000 most variable regions in i450k data were used for visualization. **(B)** When using the same set of pooled DMRs as in Fig 1E, the median DNA methylation level of PSCs is most similar to AT/RTs. **(C)** AT/RT-hyper DMRs were mostly AT/RT-unique or PSC-like, whereas MB DMRs were MB-unique or FB-like. Very few MB-hypermethylated DMRs were associated with large-scale differences in DNA methylation. Tumor type–specific DMRs (Fig 2A) were categorized based on DNA methylation levels in PSC and FB samples. The bar plot on the left shows the number of DMRs in different categories. Annotations show whether DMRs are PSC-like (P), FB-like (F), or unique (different from PSCs and FB) and whether DNA methylation changes during cell differentiation from PSC to FB. The proportion of DMRs in large-scale DNA methylation differences within annotated DMR categories is shown in blue on the right. The number of DMRs is marked in the figure. **(D)** DMR category–related DNA binding patterns revealed transcriptional regulators (TFs) involved in tumor-unique, normal cell–like, and differentiation-related regulation of DNA methylation. PRC2 subunits were enriched in the AT/RT-unique DMRs, whereas neural TFs were enriched in both AT/RT-unique and PSC-like DMRs with varying enrichment patterns. TF binding site enrichment was calculated separately for each normal cell differentiation–related DMR category (bottom). TFs were organized into the themes listed in Fig 2B. The dot is not marked when a given TF is not measured in a given GTRD category.

and are differentiation-related in a manner that underlines the similarity of MBs to FB and AT/RTs to PSCs.

To determine whether the observed DNA methylation differences were more linked to focal or large-scale regulation of DNA methylation, we calculated the proportion of DMRs involved in large-scale methylation (i.e., either hyper- or hypomethylated DMRs overpopulating certain TAD regions, Fig 1G) in each DMR category (Fig 3C, right side, Fig S10A). Notably, only 9 (8%) of MB-hyper DMRs represented large-scale DNA methylation differences, whereas 430 (50%) and 503 (55%) of AT/RT-hyper and MB-hypo DMRs, respectively, were involved in large-scale methylation (Fig 3C). Thus, MB-

hyper DMRs appear to target specific cytosines or focal areas, such as specific promoters and enhancers (consistently with their enrichment in CpG islands), whereas both focal regulation and large-scale regulation of DNA methylation are observed for AT/RT-hyper or MB-hypo regions.

## AT/RT-unique DNA methylation is associated with PRC2

We analyzed TF binding separately in the differentiation-related DMR categories presented in Fig 3C to define TFs relevant to different regulatory programs (Figs 3D and S10B). In general, the TFs

linked to SWI/SNF and H3K27ac were associated with AT/RT-unique sites (Fig 3D). Furthermore, the binding sites of EZH2 and other PRC2 members (they contribute to histone H3 lysine 27 [H3K27] methylation, so we annotated them as H3K27me TFs) were enriched only in AT/RT-unique DMRs (Fig 3D). Many of the AT/RT-unique DMRs, which are demethylated during differentiation from PSCs to FB and carry EZH2 binding site(s), were linked to genes, such as *IGSF9B*, *DOK7*, and *CAMK2B*, which are involved in dendrite growth and differentiation from neural progenitor cells to mature state (Table S3) (Okada et al, 2006; Woo et al, 2013; Wu et al, 2017). These are the key processes the neuronal BAF complex (nBAF, mammalian neuronal SWI/SNF complex) is activating during neural development (Kadoch & Crabtree, 2015). However, some of the genes harboring AT/RT-unique differentiation–demethylated DMRs with EZH2 binding, such as *TNK1*, are found to be relevant in cancer development (Hong et al, 2020) and have not been associated with nBAF or late neural development.

The SWI/SNF subunit SMARCA4, whose binding in other brain and neural cells was associated with AT/RT-hyper DMRs (Fig 2C), was enriched in non–differentiation-related AT/RT-unique DMRs (Fig 3D). SMARCA4 enrichment remained significant ($P = 0.0001$, Fisher's exact test), when only the binding sites of SMARCA4 in "other brain and neural" samples were included in the analysis. Interestingly, when using these binding sites, SMARCA4 was also significantly enriched in regions, which were similarly hypermethylated in AT/RTs and PSCs and hypomethylated during differentiation (group 3) ($P = 1 \times 10^{-7}$, Fisher's exact test). Some members of the SWI/SNF complex, such as SMARCA4 and CHD7, were also enriched in the MB-unique DMRs that are hypermethylated in MBs and non–differentiation-related (group 10). In the case of SMARCA4, these sites might represent SMARCA4 binding sites in rhabdoid tumors (significantly enriched in MB-hyper DMRs, Fig 2C), which have been DNA-methylated in an MB-specific manner, and they can potentially be also linked to the altered functions of the SWI/SNF complex in MBs in the context of *BRG1* mutation (Yi & Wu, 2018). However, fewer SWI/SNF TFs were enriched in MB-specific than AT/RT-specific hypermethylated DMRs.

Contrary to PRC2 and SWI/SNF subunits, more neural TFs (n = 12) were enriched in PSC-like AT/RT-hyper DMRs (Fig 3D). Notably, neural TFs BCHE and TFAP2A, the only neural TFs co-localizing with PRC2 subunits in AT/RT-hypermethylated sites (Fig 2D), were enriched only in AT/RT-unique DMRs (groups 1 and 2) (Fig 3D), linking them closely to PRC2-related DNA methylation.

In conclusion, our results suggest that AT/RT-specific DNA methylation is covering the sites that are bound by SWI/SNF and PRC2 complexes in neural progenitors and more differentiated neural cells and that the DNA hypermethylation in most of these sites is AT/RT-unique, not PSC-like. This aberrant DNA methylation has the potential to drive oncogenic epigenetic regulation, which is unique to AT/RTs. In addition, we detected AT/RT-hyper regions that were similarly methylated in PSCs, harbor binding sites for several neural regulators, and were originally methylated via a PRC2-independent mechanism.

## DNA methylation contributes to neural cell differentiation–related gene expression

Next, we associated DNA methylation with gene expression based on the results obtained in both microarray- and sequencing-based analysis. A total of 1,305 DE genes were detected (adjusted $P < 0.01$, logFC >= 1 in both RNA-seq and microarray data) between tumor types (Fig 4A). In the following steps, these DE genes were separately integrated with RRBS and i450k DMRs. Microarray expression datasets represented all the MB subgroups with similar subgroup proportions as in the i450k data (Tables S1 and S4) (Rathi et al, 2020). Altogether, 134 (10%) DE genes harbor an opposite change in DNA methylation (called DM-DE genes) in their genomic neighborhood in at least one tumor comparison (Fig 4B). Furthermore, 42 and 67 DM-DE genes carry DNA methylation differences in the gene promoter and the gene-annotated enhancer region, respectively (Fig 4C and D). Most of the DM-DE genes (142 of 160 genes, 89%) were detected in only one tumor comparison (Fig S11). In addition, a parallel increase/decrease in both expression and DNA methylation was detected in 130 genes, of which 70 were in AT/RT comparisons (62 in AT/RT-MB and 8 in the AT/RT-PLEX comparison, representing different genes) (Table S4).

DM-DE genes were predominantly hypermethylated in AT/RTs and underexpressed when compared to PLEXs, MBs, or both (Figs 4E, S11, and 12A and B). *CXXC5* and *TCEA3* genes were uniquely hypermethylated and down-regulated in AT/RTs, and seven genes (including neural genes *NEUROG1*, *NBEA*, and *BARHL1*) were uniquely hypomethylated and overexpressed in MBs (Fig 3E and F and S13A and B). The number of AT/RT-specific genes was partly reduced because decreased DNA methylation in both MBs and PLEXs led to increased expression in only one of them.

We identified *NEUROG1* and *NEUROD2* as DM-DE genes hypermethylated in AT/RTs (Fig 4E), and NEUROG2 and NEUROD1 as TFs whose binding sites were enriched specifically in AT/RT-hypermethylated regions (Figs 2C and S6). Because these TFs are key regulators of neural differentiation (Massari & Murre, 2000; Dixit et al, 2014), we visualized their expression together with the expression of their target genes *TBR1*, *RCOR2*, *BLHE22*, *NHLH1*, *NHLH2*, *HES6*, *DAB1*, and *CDK5R1* (Fig S14A–D). Except for NEUROG2, which had the highest expression in AT/RTs, the highest expression of all these genes was observed in MBs (Fig S14A–D). This result suggests that their induction, which should occur during neural differentiation, has been successful mainly in MBs.

## AT/RT-specific DNA methylation in genes suppressed in AT/RTs

To better understand the processes involved in AT/RT-related DNA hypermethylation, we more closely inspected the genomic neighborhoods of *CXXC5*, *TCEA3*, *NEUROG1*, *NEUROD2*, and *EBF3* genes (Fig 4G). Typically, one overlapping DMR was detected in the gene loci with both used measurement techniques (RRBS and i450k) with the exception of the *CXXC5* locus, which is located in a large-scale AT/RT-hyper region (in Chr5q31.2) and includes several DMRs (Figs 1G and 4G, S14A–D, and S15, Table S4). An AT/RT-specific DM-DE gene *CXXC5* (Fig 4G) also harbors binding sites of several neural TFs, SMARCA4 binding sites in rhabdoid tumor cells, and both PSC-like and AT/RT-unique DMRs. In addition, EP300 binding sites measured from neural cells were also found around *CXXC5*, representing possible neural SWI/SNF-EP300 target sites described earlier (Fig 4G). *TCEA3*, also an AT/RT-specific DM-DE gene, carries AT/RT-unique, non–differentiation-related DMRs and neural EZH2 binding sites at the gene promoter.

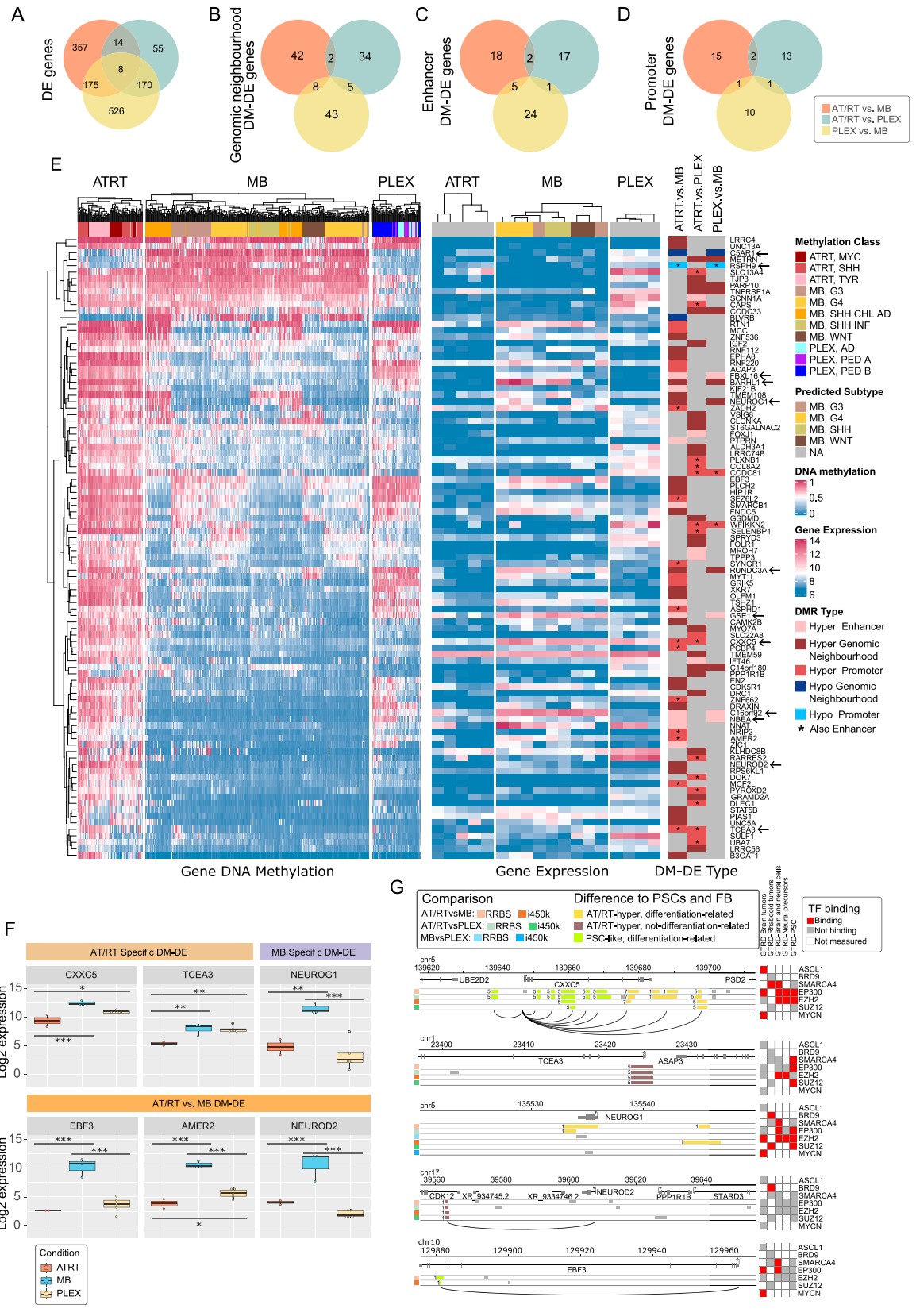

Interestingly, both *NEUROG1* and *NEUROD2* were hypermethylated and down-regulated in AT/RTs when compared to MBs in our DM-DE gene analysis and they harbor DMRs, which were hypermethylated in AT/RTs compared with PSCs and bound by BRD9 in rhabdoid tumors (Fig 4G). In addition, *NEUROG1* included EZH2 binding sites measured from several different cell types. *NEUROG1* and *NEUROD2* were also lowly expressed in all PLEX subgroups (Figs 4E and F, S13A and B, and S14A–C) and are likely to be silenced in this differentiation lineage either via alternative ways to suppress gene activity, such as H3K27me3, or by the absence of transcriptional activators in the regulatory regions. In MBs, these two TFs behaved in a complementary manner, as NEUROG1 was induced in all the other MB subgroups except in the SHH subtype, whereas NEUROD2 induction was more clearly detected in SHH and G4 subtypes (Figs 4E and S13A and B).

Finally, we wanted to investigate whether NEUROD1 binding was detected in the DMRs related to the DM-DE genes. Based on our CUT&RUN data, NEUROD1 binds in the MB cell line to the site that was connected to higher *EBF3* expression in MBs (Fig 4G, Table S4). This site was DNA-hypomethylated in MBs and is now shown to be bound by NEUROD1 in MBs, but not in AT/RTs. Interestingly, both SMARCA4 and EP300 bind to the same region in "other brain and neural samples" in the GTRD, thus suggesting that it is one of the genomic sites that these regulators are targeting during neural differentiation. Furthermore, this site has been annotated as a distal enhancer by the Encyclopedia of DNA Elements (ENCODE) project. *EBF3* has a role in neurogenesis, and it has also been shown to have tumor suppressor activity (Tao et al, 2015; Tiethof et al, 2018). Interestingly, *EBF3* has been shown to be regulated by NEUROD1 (Seo et al, 2007), and our analysis positioned NEUROD1 more specifically to its regulatory region.

Overall, our results suggest that AT/RT-related DNA methylation is involved in suppressing these genes, which can contribute to impaired further neural development characterized by AT/RTs.

# Discussion

Our analysis revealed DNA methylation patterns specific to AT/RTs and associated them with gene regulation, cell differentiation, and malignancy. AT/RT-specific features showed similar DNA methylation patterns across the tumor subgroups. Our results consistently suggest that AT/RTs remain in a less differentiated epigenetic state

compared with MBs and PLEXs, whereas the differentiation state is most advanced in MBs with focal DNA methylation of pluripotency-related TF target sites and DNA demethylation–related activation of neural genes (Hovestadt et al, 2019; Vladoiu et al, 2019; Hendrikse et al, 2022; Smith et al, 2022; Lobón-Iglesias et al, 2023). Although the characteristics of the sequencing and microarray approaches differed, the key AT/RT–specific findings align.

We showed that the binding sites of several TFs that are the drivers of neural differentiation are hypermethylated in AT/RTs, suggesting disrupted TF activity in these sites. NEUROG2 and NEUROD1 are especially interesting: they are pioneer factors inducing chromatin opening and regulate several factors affecting neural differentiation (Pataskar et al, 2016; Smith et al, 2016). Both belong to the basic helix–loop–helix (bHLH) gene family selectively binding to unmethylated DNA and are thus sensitive to DNA methylation (Yin et al, 2017). We also showed with CUT&RUN analysis that the DNA binding of NEUROD1 was decreased in AT/RTs when compared to MBs, although no major differences were observed in its protein level across studied cell lines. We also identified several regions that were hypermethylated in AT/RTs and hypomethylated in MBs, in which the NEUROD1 binding was detected solely in MBs. *NEUROG1* and *NEUROD2* were also defined as DM-DE genes with higher DNA methylation in AT/RTs than PSCs, suggesting that they could serve as differentiation bottlenecks suppressed in AT/RTs. Furthermore, for example, NEUROD/NEUROG target genes have low expression in AT/RTs, and part of them, such as OTX2 and EBF3 regulated by NEUROD1 (Seo et al, 2007; Borromeo et al, 2016), have NEUROG/NEUROD binding sites hypermethylated in AT/RTs. We also experimentally validated the binding of NEUROD1 to a distal regulatory region in the EBF3 locus in MBs, suggesting that EBF3 is one of the major downstream targets of NEUROD1 whose demethylation-associated up-regulation is relevant for the normal neural differentiation to proceed. Taken together, our results indicate the silencing of the whole differentiation cascade of genes. Several other neural genes, for example, *NBEA*, *BARHL1*, and *AMER2*, were also more methylated and less expressed in AT/RTs than MBs. Our results are supported by prior notions that neuronal differentiation factors have lowered expression in AT/RTs (Wilson et al, 2010; Erkek et al, 2019). Our study sheds light to the relevant role of DNA methylation in their suppression.

Our results suggest that EZH2, the functional subunit of PRC2 responsible for H3K27 trimethylation, contributes to AT/RT-specific DNA hypermethylation. EZH2 inhibition suppresses an embryonic stem cell–like expression pattern in SMARCB1-deficient tumors

**Figure 4. DNA methylation associated with the differential expression of genes relevant for neural differentiation and oncogenesis.**
**(A, B, C, D)** Differential DNA methylation (DM) was associated with differential gene expression (DE). Gene expression and DNA methylation patterns were studied in four contexts: differential gene expression alone (A) and DE coupled with DM in the genomic neighborhood (±200 kb from the transcription start site [TSS] within the same topologically associating domain) (B), DE coupled with DM in gene-linked enhancer (C), and DE coupled with DM in the gene promoter (2 kb upstream and 500 bp downstream from the TSS) (D). Venn diagrams show the numbers of genes behaving similarly in both sequencing and array data. Differentially expressed genes associated with differential DNA methylation (B, C, D) are called DM-DE genes. Only cases where the sign of DM change was opposite to DE were included in the figure. **(E)** DM-DE genes in AT/RT comparisons show generally high DNA methylation among AT/RTs. Sample-wise heatmaps show the levels of DNA methylation (average methylation of variable sites) and gene expression. The rightmost heatmap summarizes in which comparison the DM-DE gene was detected, what was the direction of DNA methylation change (hyper/hypo), and the genomic location of the DMR. **(F)** Expression patterns of selected DM-DE genes. *P < 0.05, **P < 0.01, ***P < 0.001. **(G)** Hypermethylated DMRs in relevant genes, which are hypermethylated and underexpressed in AT/RTs. *CXXC5* and *TCEA3* are AT/RT-specifically suppressed DM-DE genes, and *NEUROG1*, *EBF3*, and *NEUROD2* are DM-DE genes in the AT/RT-MB comparison. Distal DMRs are connected to the TSS via an arch. Oncoprint indicates which relevant TFs have binding sites in these regions in selected GTRD categories. The color of the DMR indicates whether the DMR is PSC-like and whether it is demethylated during neural cell differentiation (see Fig 1F). The number in front of the DMR indicates the k-means cluster which DMR belongs to (see Fig 1F). Gray DMRs were not included in TF binding and DMR cluster analysis as they were not AT/RT-specific.

(Terada et al, 2019), and EZH2 inactivation prevents AT/RT development and growth (Wilson et al, 2010; Alimova et al, 2013; Terada et al, 2019), making it a relevant regulator for these tumors. Because H3K27me3 levels and other studied histone marks are mainly depleted in AT/RTs (Erkek et al, 2019), the elevated DNA methylation is likely to play a role in AT/RT malignancy. PRC2 marks many genes that are methylated during cell differentiation from PSCs to neural precursor cells (Cedar & Bergman, 2009). EZH2 has been reported to interact with DNA methyltransferases and even to directly control DNA methylation (Viré et al, 2006), which could partly explain AT/RT-unique DNA hypermethylation we observed. In rhabdoid tumor cells, DNA methylation is decreased and PRC2 members are concomitantly released from the p16 locus upon SMARCB1 re-expression (Kia et al, 2008), also suggesting an interplay between PRC2, SWI/SNF, and DNA methylation. In our results, many of the genes linked to EZH2-bound sites with AT/RT-unique DNA hypermethylation are related to late neural differentiation and nBAF. Most of these genes were lowly expressed also in MBs and/or PLEXs. However, we also detected EZH2 binding sites in AT/RT-unique DMRs in DE genes such as *NEUROG1* and *TCEA3*, which are shown to be down-regulated in cancer and are relevant for early development (Park et al, 2013). In addition, there are AT/RT-unique DMRs lacking EZH2 binding sites in relevant genes, such as *NEUROD2*. This might be due to, for example, incomplete EZH2 binding data (not all cell types are covered in the GTRD) or EZH2-independent regulation. As only a subpopulation of *SMARCB1*-deficient tumors responds to EZH2 inhibition in clinical trials (Italiano et al, 2018; Gounder et al, 2020), the functional role of EZH2 appears to be patient-specific and at least partly decreased after oncogenesis. The balance between EZH2-induced H3K27me3 and DNA methylation is still unclear, and constant EZH2 activity is most likely not needed for sustaining the DNA hypermethylation, which appears to have a significant role in AT/RTs.

PSC-like DNA methylation and gene expression patterns have been shown to exist in AT/RT samples (Terada et al, 2019). We showed that these PSC-like regions are less methylated in MBs and PLEXs and differentiation-related, uncovering their functional role and strengthening their relevance. Our results suggest that targeted DNA demethylation is one of the key processes during neural cell differentiation, which exposes the enhancers and promoters of neural genes for controlled gene activation. This finding suggests that these PSC-like sites remain methylated, and thus inaccessible, in AT/RTs, which maintains AT/RTs in a low differentiation state and promotes malignancy. We also showed that many of these regions involve large-scale DNA methylation differences, implying larger changes in the chromatin. As the SWI/SNF complex has been shown to guide DNA demethylation during cell differentiation (Yildirim et al, 2011; Sepulveda et al, 2017), the PSC-like DNA methylation can reflect SWI/SNF complex's inability to target DNA demethylation in the absence of *SMARCB1*. AT/RT-unique DNA methylation, which appears to also silence key neural regulators and other neural genes, can also halt AT/RT cells at a specific point in the developmental trajectory by suppressing the expression of key neural regulators, such as *NEUROG1* and *NEUROD2*, and prohibit the downstream DNA demethylation steps necessary for further neural development. The contribution of both mechanisms is supported by the enrichment of SMARCA4 binding sites in neural cells in both AT/RT-unique and PSC-like AT/RT-hyper DMRs. Overall, our results

suggest that DNA methylation leads to halted neural cell differentiation and drives malignancy in AT/RTs.

It is known that SMARCB1 loss needs to happen during certain developmental stages for rhabdoid tumor development, whereas mainly lymphomas and benign tumors are formed at later phases (Han et al, 2016; Bracken et al, 2019; Parisian et al, 2020). Based on our results, the temporal relationship to the DNA demethylation might be one of the key determinants of the cell fate after SMARCB1 inactivation. This is supported by recent findings that AT/RTs respond well to DNA methyltransferase inhibition both in vitro and in vivo (Steinbügl et al, 2021; Graf et al, 2022). Further research will show if this or similar therapeutic approaches could benefit patients suffering from this devastating disease.

# Materials and Methods

### Sequencing cohort

Tumor samples analyzed with next-generation sequencing were collected for this study from 10 patients operated at Tampere University Hospital. The study was performed in line with the principles of the Declaration of Helsinki. Approval for this study was granted by the Regional Ethics Committee of Tampere University Hospital (decisions: R13050, date 9.4.2013, R14024, date 28.3.2014, and R07042, date 20.9.2017) and Valvira (decision V/78697/2017). Written consent was obtained from patients except for 1 MB and 2 PLEX samples, which were collected as part of the different cohorts. An experienced neuropathologist evaluated the tumor samples and determined the histopathological type and grade according to the criteria presented by the WHO 2016 (Louis et al, 2016).

### Cell culture

AT/RT cell lines CRL-3020, CRL-3036, and CRL-3038 and a medulloblastoma cell line CRL-3021 were cultured in DMEM/Hepes with 2% B27, 1% penicillin–streptomycin, 20 ng/ml FGF, and 20 ng/ml EGF. All the cells were cultured in an incubator at 37°C with 5% $CO_2$.

### RRBS

DNA was isolated from the frozen samples and cell lines using a QIAamp DNA Mini kit (QIAGEN) with RNAse treatment and from the FFPE sample with a turXTRAC FFPE DNA kit (Covaris). Library construction and sequencing were performed at the Beijing Genomics Institute (BGI), Hong Kong. Samples were sequenced using an Illumina HiSeq 2000 technology. Paired-end sequencing of 50 bp was used. The data goal was 5 Gb. Cell lines were sequenced at Novogene, with Illumina NovaSeq paired-end 150 bp. The data goal was 15 Gb.

### RNA sequencing

RNA was isolated using a mirVana isolation kit (Invitrogen). Library construction and sequencing were performed at Novogene, Hong Kong. Samples were sequenced using an Illumina HiSeq 2000

technology. Paired-end sequencing of 150 bp was used. The data goal was 20 million raw reads per sample.

## Western blotting

Samples from cell lines were scraped on ice with RIPA cell lysis buffer (Thermo Fisher Scientific) supplemented with protease inhibitors (Roche). Proteins were separated by a Ready Gel precast gel (Bio-Rad) and transferred onto a nitrocellulose membrane. Blocking was done with 5% milk, 1 h at RT. Primary antibody incubations were performed overnight at 4° for both polyclonal NEUROD1 (1:500, PA5-47381; Invitrogen) and monoclonal $\beta$-tubulin (1:20,000, SAP.4G5). Secondary antibodies (polyclonal rabbit anti-mouse immunoglobulins/HRP [1:1,000, P0260; Dako] and polyclonal rabbit anti-goat immunoglobulins/HRP [1:2,000, P0160; Dako]) were incubated for 1 h at RT. All antibody dilutions were made to 5% milk. Proteins were detected by enhanced chemiluminescence (Bio-Rad).

## CUT&RUN sequencing

Each cell line was harvested by removing the media with cells from flasks and centrifuging the media at 175$g$ for 7 min. The supernatant was removed, and cells were washed once with PBS followed by centrifugation at 175$g$ for 7 min. Cells were counted, and 100,000 cells were collected for each reaction. CUT&RUN was performed using the Cell Signaling Technology CUT&RUN Assay kit (#86652). The experiment was done as described in the kit's protocol. NEUROD1 antibody #PA5-47381 from Invitrogen was used. DNA concentrations were measured using Qubit 2.0. Library construction was done using DNA Library Prep Kit for Illumina Systems (ChIP-seq, CUT&RUN; #56795). Sequencing was performed at Novogene, UK. Samples were sequenced using an Illumina NovaSeq X technology. Paired-end sequencing of 150 bp was used. The data goal was 15 million raw reads per sample.

TrimGalore! version 0.6.7 was used to trim adapter reads. Reads with a length of less than 25 bp were filtered out. Reads were aligned to hg38 using bowtie2 version 2.4.5 and the −dovetail flag. Duplicated reads were removed using Picard MarkDuplicates version 2.27.1. Regions overlapping with the regions described by Nordin et al (2023) were removed using bedtools intersect version 2.29.1. Afterward, the reads with a length shorter or equal to 120 bp were extracted using samtools view (version 1.8). MACS3 version 3.0.0.bl was used to call the peaks for the fragments with the indicated length. Peaks located outside of the canonical chromosomes and on chrX or chrY were filtered out. Peaks that overlapped with peaks in the negative control samples (input DNA) were removed using bedtools intersect version 2.29.1. Replicates were unified by keeping peak regions called in both replicates. For all remaining peaks, the fasta sequence was extracted using bedtools getfasta. The NEUROD1 motif was downloaded from JASPAR 2020 (Fornes et al, 2020), and all peaks were scanned for the motif using FIMO (part of MEME suite v. 5.5.5) (Grant et al, 2011), keeping all peaks with a detected motif ($P < 0.001$). The peaks obtained with this method were combined with peaks overlapping with GTRD v.19.04 NEUROD1 binding sites and overlapped with obtained DMRs in R, using the GenomicRanges package (version 1.34.0).

## Preprocessing and analysis of DNA methylation datasets

### i450k data

The full CNS tumor reference cohort GSE90496 established by Capper et al was downloaded from Gene Expression Omnibus (GEO) (Edgar et al, 2002; Barrett et al, 2013), filtered, batch effect–corrected, and normalized as described in their study (Capper et al, 2018), resulting in an initial set of 428799 probes and 2801 samples. The Capper method includes the following probe filters: (1) removal of probes targeting the sex chromosomes, (2) removal of probes containing a single-nucleotide polymorphism (dbSNP132 Common) within five base pairs of and including the targeted CpG site, (3) probes not mapping uniquely to the human reference genome (hg19) allowing for one mismatch, and (4) probes not included on the Illumina EPIC array. Supplementary clinical annotations were also mapped onto the samples. AT/RT, MB, PLEX, and CONTR samples, excluding the control subtype "INFLAM" and all samples without location information, were then extracted (N = 584).

To detect DMRs between tumor types, AT/RT-MB, AT/RT-PLEX, and PLEX-MB comparisons were conducted, and each cancer was also tested against the CONTR set. Additional probe filtering (removing probes not mapping to the hg38 genome and the cross-hybridizing probes) followed by DMR calling for the remaining 396700 probes was performed with DMRcate v1.18.0, switching the hg19 coordinates into hg38 with rtracklayer v1.42.2 (Lawrence et al, 2009; Peters et al, 2015; R Core Team, 2019). The beta cutoff threshold was set to 0.05, and FDR < 0.05, but otherwise the default settings were used. Location information was included in the linear model to adjust for biological confounders. To reduce normal cell effects, the DMRs of each main comparison (e.g., AT/RT-MB) were overlapped with the control DMRs (e.g., AT/RT-CONTR, MB-CONTR) using the subsetByOverlaps function. The resulting regions were reduced by taking only the intersecting regions between the main comparison and at least one of the control comparisons (Lawrence et al, 2013). Based on the visual inspection of the regions using heatmaps, the final differential methylation criteria used to define the DMRs most relevant to further analyses were increased from 0.05 to 0.20.

DMRs were annotated to annotatr v1.8.0 pre-built hg38 genic regions, CpG regions, and FANTOM5 enhancers (Lizio et al, 2015; Cavalcante & Sartor, 2017). In addition, the pre-built promoter annotations (1 kb upstream from the transcription start site [TSS]) were used to generate broader promoters (2 kb upstream and 500 bp downstream from the TSS), and genomic neighborhoods spanning 200 kb in both directions from the TSS. Moreover, enhancer information from GeneHancer was used, selecting only the confirmed Promoter/Enhancer records with a confidence score >= 5 (Fishilevich et al, 2017).

### RRBS data

Quality control, adapter trimming, and MspI restriction site trimming were performed using the Babraham Bioinformatics Group preprocessing tools FastQC v0.11.7 and Trim Galore v0.5.0. Reads were queried against bisulfite-converted reference genomes of potential contaminants with FastQ Screen v0.13.0 to check the origin of the libraries (Wingett & Andrews, 2018). Mapping against hg38 (UCSC) reference and mapping quality control without duplicate removal were performed using Bismark v0.19.1 (Krueger &

**Life Science Alliance**

Andrews, 2011). The quality information of all these steps was summarized over the dataset using MultiQC (Ewels et al, 2016).

Methylation percentage calling and differential DNA methylation analysis were performed using methylKit v1.8.1 (Akalin et al, 2012; R Core Team, 2019). CpGs with coverage of less than 10 reads or hitting into scaffolds, sex chromosomes, or mitochondrial DNA were removed. The CpGs were tiled into 1,000-bp non-overlapping regions, that is, tiles, which were used for DMR calling with overdispersion correction. Regions with DNA methylation percentage difference greater than 25% and q value (FDR) smaller than 0.05 were extracted and then annotated similarly as i450k data.

### Dimensionality reduction

The i450k beta values for all AT/RT, MB, and PLEX samples were aggregated on RRBS tiles based on lift-over hg38 probe coordinates. This step was performed by taking the complete overlaps and then calculating the mean beta value for probes hitting each tile (Lawrence et al, 2013). The scales in the dataset were unified, and the 10,000 most varying regions were used as inputs for dimensionality reduction algorithms.

### K-means clustering

The i450k DMRs (q < 0.05, meth. fold change >= 0.20, three comparisons) were combined into a single pool of regions (n = 3,780). These were used to generate a DNA methylation matrix such that the mean of the probes hitting each region was used to represent the DNA methylation level for the region in each sample. K-means clustering was calculated with the cluster amounts from 1 to 25, each time using 25 iterations and 50 starting points. The optimal number of clusters was estimated with the Akaike information criterion, resulting in k = 10. DMR behavior in clusters was visually inspected using heatmaps. Moreover, the regions belonging to each cluster were aggregated using the median to summarize the average DNA methylation patterns in the sample types. The general DNA methylation–level differences between the subtypes among these regions were also visualized as a violin plot and estimated using two-sided Wilcoxon rank sum tests.

### TF binding analyses

### Preliminary statistical testing

TFBS enrichment analysis was performed using one-sided Fisher's exact test (FET). TFBS data originating from ChIP-seq measurements were obtained from GTRD v.19.04 metaclusters (Yevshin et al, 2019). Before testing, DMRs with beta fold change >= 0.25 were filtered by overlap operations between the comparisons to obtain the regions specific to each cancer. Hypermethylated and hypomethylated regions were tested separately. For the i450k data, the background set was generated from previously filtered 396700 probes, and for the RRBS data, the methylKit tiles were used as the background.

When performing the tests with i450k data, the hg38 coordinates of each 50-bp probe overlapping with DMRs were first obtained. These were extended to 500 bp in both directions, and the DNA methylation fold change of the original DMR was used to represent the methylation difference in each extended probe. Regions were considered cancer-specific if they overlapped with at least 600 bp

between comparisons, and with RRBS, because of the non-overlapping nature of the DMRs, the exact overlaps could be used.

FET P-values were adjusted by the Benjamini–Hochberg (BH) correction (Benjamini & Hochberg, 1995). TFs with adjusted P-values smaller than 0.05 were considered significant. Sets of validated TFs with significant enrichment in the analogous DMR sets, that is, in hypermethylated or hypomethylated cancer regions in both i450k and RRBS experiments, were constructed. The functions of the enriched TFs were manually curated, based on which 11 recurrently detected TF functions and families were chosen as TF themes. Then, the possible link between each TF and TF theme was studied again with a manual literature search in a consistent manner.

TFBS enrichment analysis of k-means clustering results was carried out using a similar method to the above (one-sided Fisher's exact test, BH correction). Cancer-specific DMRs from 450k data were assigned to the clusters, and TFBS enrichment was tested for all GTRD TFs with all clusters, cancer types, and DNA methylation changes (hyper/hypo) by examining all autosomal i450k probes and associating them with TF binding and DMRs to create the contingency table.

### Customized TFBS enrichment analysis with categorized GTRD data

GTRD annotations for cell types linked to a given TFBS were further examined and manually divided into 11 groups (Table S2). The GTRD was then categorized based on the terms of the groups, and these groups were used as input material for the FET to replace the full GTRD.

### Studying average DNA methylation levels at binding sites

DNA methylation of the relevant i450k probes near the binding sites of the validated TFs (maximum gap 500 bp) was visualized as violin plots. The probes were selected by pooling the cancer regions and then finding complete overlaps with the hg38 probe coordinates. The beta values over each tumor subclass were aggregated probewise using the mean.

### TFBS co-localization

The co-localization of validated TFs was studied with TFBSs hitting target areas (hypermethylated AT/RT regions, hypermethylated MB regions, hypomethylated MB regions, or hypomethylated PLEX regions). TFs were tested pairwise against each other to construct four-field matrices for each case. Regions with (1) no binding site from either TF, (2) binding sites from both TFs, and (3–4) binding sites from one TF but not the other were counted, and the significance was tested using one-sided FET and stored as a symmetric TF matrix of P-values. These P-value matrices were corrected using the BH method. Adjusted P-values were subset by the TF names of the validation group (AT/RThyper-TFs, MBhyper-TFs, MB-hypoTFs, PLEXhypoTFs) and visualized using the ComplexHeatmap package (Gu et al, 2016).

### Differential expression analysis for expression datasets and integration with DNA methylation data

### Array datasets

Normalized expression arrays of three public datasets were downloaded from GEO: Illumina HumanHT-12 V3.0 (Henriquez et al,

2013) (accession GSE42658) and two Affymetrix Human Genome U133 Plus 2.0 sets (Birks et al, 2013) (accession GSE35493 and [Sturm et al, 2016] accession GSE73038). The Illumina set contains measurements of all three cancers, and Affymetrix sets provide additional support for the AT/RT-MB comparison. No normal brain, PSC, or FB samples were included in the gene expression analysis, so differentially expressed genes were solely called between tumor types. All the array expression datasets were first analyzed separately. Before DE analysis with limma, the expression matrices were log$_2$-transformed if that had not been performed by the normalization method (Ritchie et al, 2015). Probes were annotated into symbolic gene names, and the mean of all the probes mapped to the same gene was chosen to represent the expression values. Genes with BH adjusted P-values smaller than 0.05 and log$_2$-transformed fold changes (absolute change) of at least 1 were considered significant. Only those genes were selected that were differentially expressed in GSE42658, or alternatively in both GSE35493 and GSE73038.

### RNA-seq data

After sequencing data QC with FastQC v0.11.7 (https://www.bioinformatics.babraham.ac.uk), reads were mapped and quantified against hg38 by kallisto v0.44.0 (Bray et al, 2016). The kallisto transcript abundance was transformed to gene-level counts and scaled to library size using tximport v.1.10.1. DE analysis was then performed using DESeq2 v1.22.2, with default settings (Love et al, 2014). Genes with BH adjusted P-values smaller than 0.05 and log$_2$-transformed fold changes (absolute change) of at least 1 were considered significant.

### Integration

The DE results were merged with the annotated DMRs by symbolic gene names to find cases where the direction of the DNA methylation change in a regulatory region (i.e., promoter, enhancer, genomic neighborhood) was opposite to the gene expression change. A similar analysis was also done for the genes with up/down-regulation of both DNA methylation and gene expression. GEO expression array results were integrated with i450k DMRs, and RNA-seq–based results were integrated with RRBS DMRs from matched cases to provide data for validation. Validation was performed by extracting the genes that showed the similar behavior in both approaches. Because annotatr uses transcript-level annotations, the validated hits, for example, for promoters, represent DE genes that show regulatory DNA methylation in at least one of its transcript-specific promoters in both datasets. Additional filtering and validation steps were performed for the enhancers and genomic neighborhoods, described later. The results of the experiments were summarized using Venn diagrams and an oncoprint-like visualization using the Complex-Heatmap package (Gu et al, 2016).

A recent regulatory genomics workflow was adapted to integrate DMRs and DE genes with pre-calculated promoter–enhancer annotations provided by the FANTOM5 database (Lizio et al, 2015; Ferrero, 2018). DE-DMR pairs in genomic neighborhoods were queried between i450k and RRBS experiments with a maximum gap of 5 kb, and the regions inside this "validation distance" with a similar DNA methylation direction were extracted. Moreover, to ensure that a given DMR can spatially interact with the TSS of a given gene, the results were studied with TAD information from Hi-C experiments (Dixon et al, 2012; Wang et al, 2018) (http://3dgenome.fsm.northwestern.edu/). The region between the DMR and the TSS of each record overlapped with TADs originating from five cell types: SKNDZ, Cortex DLPFC, Hippocampus, H1-ESC, and H1-NPC (Wang et al, 2018). Genes without any TAD boundaries in any of their transcript TSS-DMR pairs were extracted.

### Linking DMRs to TADs for large-scale DNA methylation analysis

DMRs were overlapped with *Cortex DLPFC* and H1-NPC TADs using Bedtools GroupBy v2.25.0 and in-house scripts to calculate hyper- and hypomethylated records in each TAD (Quinlan & Hall, 2010; Wang et al, 2018). The input material was either a broadened (±500 bp) probe extracted from the i450k DMRs or the methylKit DMRs of the RRBS analysis. The resulting TADs for each comparison were filtered such that at least five similarly directed probes/tiles, each with a DNA methylation difference of at least 25% with both the i450k and RRBS approach, should hit the TAD. Simultaneously, less than 1/10 of the hits can undergo DNA methylation changes in the opposite direction. Moreover, the distance between the first and last probe or tile in the TAD had to be over 50 kb. The filtered TADs from different cell lines were pooled and reduced to produce single TAD coordinates for each comparison and then visualized with karyoploteR (Gel & Serra, 2017).

### Comparing brain tumors with ESCs and primary fetal brain

Two PSC samples, nine primary FB samples (GSE116754) (Colunga et al, 2019), and 25 PSC samples (GSE31848) (Nazor et al, 2012) were downloaded from GEO and preprocessed using the Capper et al data (Capper et al, 2018). Beta-value density plots and tSNE clustering for the 10,000 most varying probes were used to check that samples were separated based on the tumor and normal sample type. The DMRs were called with DMRcate by comparing AT/RT, MB, and PLEX samples against PSC and FB cells. In addition, PSC samples were compared with FB.

From these seven comparisons, hypermethylated and hypomethylated DMRs were extracted (q < 0.05, abs. average beta fold change >= 0.25, and length between 100 and 5,000 bp). The region lists were overlapped with the same cancer-specific regions extracted for TFBS analysis (min 1 bp overlap). As a result, the experiment was summarized in a "developmental DMR matrix" of 2,301 non-overlapping regions showing which of the comparisons or cancers each region overlapped. The resulting region sets were categorized based on the differences detected in tumor–tumor, tumor–normal, and PSC-FB comparisons. TFBS enrichment analysis was performed for these categories by extracting and extending the probes hitting the regions and testing them against the GTRD using one-sided FET, as described.

### Code availability

The most relevant codes used in this study are available in https://github.com/CRI-group/DNA_meth_ATRT.

## Data Availability

The CUT&RUN, RNA-sequencing and RRBS data generated in this study have been submitted to the NCBI Gene Expression Omnibus database (https://www.ncbi.nlm.nih.gov/geo/) under the accession number GSE197569.

### Ethics statement

The use of the clinical material was approved by the Ethical Committee of the Tampere University Hospital (approval numbers R13050, R07042, and R14024).

## Supplementary Information

## Acknowledgements

We would like to acknowledge Mrs. Paula Kosonen, Mrs. Päivi Martikainen, Mrs. Marika Vähä-Jaakkola, Mrs. Marja Pirinen, Mrs. Sari Toivola, Mrs. Hanna Selin, Mrs Marita Nieminen, Mrs Leena Jalonen, and Mrs Satu Ranta for sample handling and logistics, Mrs. Aliisa Tiihonen and Mrs Maria Pere for scientific input, Mrs. Suvi Lehtipuro for preprocessing of i450k data, and Mr. Tomi Häkkinen for technical support. Personnel at Tampere University Hospital and Fimlab Laboratories Ltd. are acknowledged for their contribution to sample collection. We are grateful to them and the patients for permitting the analysis of patient material. The study was financially supported by the Academy of Finland (#312043 [M Nykter], #310829 [M Nykter], and #333545 [KJ Rautajoki]), Cancer Foundation Finland (M Nykter, KJ Rautajoki), Sigrid Jusélius Foundation (M Nykter, KJ Rautajoki), Emil Aaltonen Foundation (KJ Rautajoki), Finnish Cancer Institute (M Nykter), Competitive State Research Financing of the Expert Responsibility Area of Tampere University Hospital (M Nykter, KJ Rautajoki), Väre Research foundation (K Nordfors, KJ Rautajoki), Aamu Pediatric Cancer Foundation (KJ Rautajoki), Finnish Pediatric Research Foundation (K Nordfors), and Finnish Cultural Foundation (J Uusi-Mäkelä). We acknowledge the CSC—IT Centre for Science, Finland, for providing computational resources and the Biocenter Finland (BF) and Tampere Genomics Facility for the service.

### Author Contributions

M Pekkarinen: formal analysis, investigation, visualization, methodology, and writing—original draft, review, and editing.
K Nordfors: conceptualization, resources, data curation, and writing—original draft.
J Uusi-Mäkelä: validation, investigation, visualization, and writing—original draft, review, and editing.
V Kytölä: formal analysis, investigation, visualization, methodology, and writing—original draft.
A Hartewig: resources, validation, investigation, visualization, and writing—review and editing.
L Huhtala: validation, investigation, visualization, and writing—review and editing.
M Rauhala: resources, validation, investigation, visualization, and writing—review and editing.
H Urhonen: investigation and visualization.
S Häyrynen: validation and investigation.
E Afyounian: validation and investigation.
O Yli-Harja: investigation.
W Zhang: resources and investigation.
P Helen: resources and investigation.
O Lohi: resources and investigation.
H Haapasalo: resources and investigation.
J Haapasalo: resources and investigation.
M Nykter: data curation, validation, and investigation.
J Kesseli: conceptualization, resources, data curation, funding acquisition, validation, investigation, visualization, project administration, and writing—original draft, review, and editing.
KJ Rautajoki: conceptualization, resources, funding acquisition, validation, investigation, visualization, project administration, and writing—original draft, review, and editing.

### Conflict of Interest Statement

The authors declare that they have no conflict of interest.

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
