## [Reviewer comments · Life Science Alliance]

Life Science Alliance

Aberrant DNA methylation distorts developmental trajectories in atypical teratoid/rhabdoid tumors

Meeri Pekkarinen, Kristiina Nordfors, Joonas Uusi-Makela, Ville Kytola, Anja Hartewig, Laura Huhtala, Minna Rauhala, Henna Urhonen, Sergei Hayrynen, Ebrahim Afyounian, Olli Yli-Harja, Wei Zhang, Pauli Helen, Olli Lohi, Hannu Haapasalo, Joonas Haapasalo, Matti Nykter, Juha Kesseli, and Kirsi Rautajoki

DOI: <https://doi.org/N/A>

Corresponding author(s): *Kirsi Rautajoki, Tampere University*

Review Timeline:

Submission Date:	2023-04-11
Editorial Decision:	2023-05-26
Revision Received:	2024-01-31
Editorial Decision:	2024-02-28
Revision Received:	2024-03-06
Accepted:	2024-03-06

Transaction Report:

May 26, 2023

Re: Life Science Alliance manuscript #LSA-2023-02088-T

Kirsi J. Rautajoki
Tampere University

Dear Dr. Rautajoki,

Thank you for submitting your manuscript entitled "Aberrant DNA methylation distorts developmental trajectories in atypical teratoid/rhabdoid tumors" to Life Science Alliance. The manuscript was assessed by expert reviewers, whose comments are appended to this letter. We invite you to submit a revised manuscript addressing the Reviewer comments.

Thank you for this interesting contribution to Life Science Alliance. We are looking forward to receiving your revised manuscript.

Sincerely,

B. MANUSCRIPT ORGANIZATION AND FORMATTING:

Reviewer #1 (Comments to the Authors (Required)):

Pekkarinen et al have submitted a manuscript on the role of DNA methylation in the control of gene expression in atypical teratoid rhabdoid tumor (ATRT), a lethal disease of the CNS that occurs in very young children. The mechanisms controlling this disease are reasonably well understood at this point, but there is certainly a need to better define epigenetic mechanisms controlling tumorigenesis. The manuscript is written reasonably well, though the numerous comparisons made in the paper are difficult to follow at times, and the manuscript is quite lengthy. I do not think language-specific editing is required. I have no concerns regarding conflicts of interest. Given the extensive reliance on normalization across studies, I would recommend the methods be thoroughly reviewed by a bioinformatician if the paper is considered for publication.

In summary, the authors present an analysis of largely previously published DNA methylation and gene expression (microarray and RNA-Seq) in a set of ATRTs as well as medulloblastomas and choroid plexus tumors (pathology not defined) in order to define an ATRT-specific methylation signature. The authors use a bioinformatics analyses that leverage publicly available tissue and cell-line ChIP-Seq data to infer the impact of this methylation signature on transcription factor function, concluding that differentially methylated regions in ATRT block the activity of transcription factors and chromatin remodelers, which itself locks ATRT precursor cells into a more 'primitive' cell state.

Overall, the submitted manuscript contributes little new data to the literature (10 samples that underwent bisulfite sequencing and RNA sequencing). While the authors' concept of using DNA methylation signatures to predict how SMARCB1 loss impacts lineage-specific transcription factor function is interesting, the analyses herein rely largely on circumstantial evidence. No experiments were performed to validate predictions made by bioinformatics analyses. More concerningly, there is no clear rationale for the comparisons made: there is no biological reason to use medulloblastoma or any choroid plexus tumor as a control tissue for ATRT. In this setting, I view the study as fundamentally flawed, but some of these could be addressed with substantial changes to the manuscript.

From an experimental standpoint, I think the following would be necessary to sufficiently substantiate the authors' claims. There is a prevailing assumption throughout the manuscript that DNA methylation at transcription factor binding sites precludes their function in ATRT, and that differentially methylated regions control distinct transcription factor functions in ATRT and medulloblastoma, but there are no experimental data to support this conclusion in the manuscript. The authors focus on NEUROG1 and NEUROD2 as potential drivers of neural differentiation that are unable to function in ATRT due to their target sites being methylated, but no experimental data are presented to support this. I would recommend at least performing ChIP-Seq (or CUT&RUN) to these transcription factors and important enhancer-associated epigenetic marks (H3K27ac, H3K4me1, H3K4me3 and H3K27me3) as well as DNA methylation sequencing in either an experimental model of ATRT or in ATRT specimens and a control tissue (ie, normal brain or fetal brain, if available). I think the former approach would be both more feasible and mechanistically more convincing.

Other major edits:

- Please provide a sound rationale for using medulloblastoma and choroid plexus tumors as control groups in the Results or Methods section. At the moment, this appears to be a comparison of convenience.
- For GTRD use, what restrictions were placed on the tissues / cell lines used in the analysis? TF binding sites are lineage-specific, so this has implications for how we interpret the output from this computational tool. For instance, it does not make sense to consider a previously validated EZH2 binding site in a pancreatic cancer to be considered a 'lost' EZH2 binding site in ATRT.
- The use of a manual literature search to annotate enriched TFs and chromatin remodelers into different 'themes' is vague and subject to selection bias. The authors provide no rationale as to why the selected proteins were chosen aside from their known roles in cell differentiation.
- There are numerous instances throughout the manuscript where it is not clear whether the authors are referencing DNA methylation or H3K27 trimethylation. The authors should make all references to methylation clear, especially when discussing polycomb complexes.
- It is often difficult to follow the authors' references to which data they are analyzing. The comparisons made between ATRT and other tumor types / tissues seem arbitrary, and descriptions of findings often do not follow a specific line of inquiry. For instance, page 10, line 22: What is meant by the "split GTRD-data"?

Minor edits:

- The studies by Torchia and colleagues outlining the molecular subgroups of ATRT should be referenced in the manuscript, as these were seminal studies in defining our molecular understanding of ATRT.

- Figure 1

o Why were the 10 matched RRBS / RNA-Seq samples included in the analysis? These are considered outliers in the tSNE plot?

o Did the authors consider restricting their analysis to only AT/RT vs. fetal brain and PSCs? And restricting the DMR analysis to AT/RT subgroups only? This would be more likely to bring out subgroup-related differences.

- Figure 2:

o Only 'cancer-specific' DMRs were considered. How was this defined, and why weren't other DMRs considered to be of interest?

o The labeling of 'TFs' is misleading in Figure 3C; many of the labeled proteins / complexes are not TFs but rather chromatin remodeling complexes, histone deacetylases or their subunits (ie, ARID2, BRD9, SMARCA4, EZH2, EP300).

o P. 9, line 44, "Together these findings suggest that DNA methylation reduces the DNA binding of neural TFs, and thus neural differentiation, in AT/RT, 2 whereas a more advanced neural differentiation is allowed in MB." This statement is not grounded in the data presented in the paper. No data regarding DNA binding of any of the listed proteins were presented in Figure 2.

- Figure 3:

o What is the rationale for considering large-scale DMRs separately, and how are these defined?

o "In conclusion, our results suggest that AT/RT-specific methylation is linked to malfunctioning 11 SWI/SNF and PRC2 complexes" - This conclusion is not supported by the data presented in the manuscript, nor is this necessarily a novel concept. The authors frequently discuss 'methylation' in relation to PRC2 function in this section - are they referring to DNA or histone methylation, as polycomb complexes are generally thought to promote H3K27 trimethylation and are not known to have a DNA methylation function. Please clarify.

- Figure 4

o The authors indicate methylation at predicted enhancers results in altered expression of target genes, but no experimental data are shown in this regard. Are there data to support the notion that DNA methylation alters enhancer activity? Or are the authors confusing DNA methylation with H3K27 trimethylation? The authors themselves acknowledge there is a mechanistic gap in this logic (page 18, line 27: "The balance between EZH2-induced H3K27me3 and DNA methylation is still unclear") ("Since 9 H3K27me3 levels and other studied histone marks are mainly depleted in AT/RTs (Erkek et al. 2019), the elevated DNA methylation is likely to play a role in AT/RT malignancy.")

Reviewer #2 (Comments to the Authors (Required)):

This study aimed to investigate the distinctions between atypical teratoid/rhabdoid tumors (AT/RTs), medulloblastomas (MBs), and choroid plexus tumors (PLEXs), which are brain tumors found in infants. These tumors are characterized by a low level of differentiation, leading to various cancer-related characteristics. The authors reanalyzed existing DNA methylation and gene expression datasets of AT/RTs, MBs, and PLEXs, along with samples from pluripotent stem cells (PSCs) and fetal brains. They examined DNA methylation patterns, associated DMRs with transcription factors (TFs) occupancy, and correlated with changes in gene expression to gain insights into the dissimilarities among these tumor types. In addition, they collected RRBS and RNA-seq data from 10 patients and integrated them into the analyses. Throughout the manuscript, the authors primarily focused on the results observed in hypermethylated differentially methylated regions (DMRs) in AT/RTs. Cross-comparison between AT/RTs, MBs, and PLEXs would interest the brain cancer field as well.

Overall, the bioinformatic analyses in the manuscript are well-executed and the figures are presented effectively. However, some concerns need to be addressed before accepting the manuscript. These main points are outlined below:

1. Lack of full description of analyses and results in the main text: The authors should provide more information about their analyses and results in the main text. This would help readers better understand the methodologies employed and the outcomes obtained. Additionally, the authors should provide a clearer rationale and implications in the results section. For example, they could elaborate on the cell type classification and its implications for the analyses conducted using the Gene Transcription Regulation Database (GTRD). Similarly, the authors should explain the rationale and implications behind conducting analyses related to PSC/FB and differentiation. Some of the rationale and impact are currently discussed in the manuscript's discussion section - it would be more appropriate to move some into the results section.

2. Lack of sufficient description of results related to MBs and PLEXs: Although the authors focused primarily on AT/RTs and hypermethylated differentially methylated regions (DMRs) in AT/RTs throughout the manuscript, there are also notable changes observed in medulloblastomas (MBs) (as evident in Figure 2B-C and Figure 3C-D). The authors need to provide more details about these results. It is essential to provide a comprehensive understanding of the findings.

Specifically, in Figure 2 - to connect DMRs to biological functions, the authors focused on TFs motifs analyses in DMRs. Using ChIP-seq data depository and literature search, they sorted out TFs linked to neural differentiation and neural cell-related TFs uniquely enriched in AT/RT-hyper DMRs.

The authors overall concluded, "Together these findings suggest that DNA methylation reduces the DNA binding of neural TFs, and thus neural differentiation, in AT/RT, whereas a more advanced neural differentiation is allowed in MB." However, the analyses showed DNA methylation increases at pluripotency TF binding sites in MBs (Figure 2C), implying disrupted pluripotency in MBs. The authors should comment on the overall picture of what occurs in MBs.

Similarly, the statement "Out of these, 10 (67%) were enriched in AT/RT-hyper sites, MYCN in AT/RT-hypo, and only ZEB1 in MB-hyper regions". The authors did not mention the enrichment of REST and SIN3A in MB-hyper regions. Why?

This kind of interpretation, which solely focuses on AT/RT hyper DMRs, continued throughout the study. In Figure 3D, the

authors present DMR meta-analyses primarily centered around AT/RT hyper DMRs. However, a similar defect is observed in regions occupied by the SWI/SNF and PRC2 complexes in both AT/RT hyper and MB hyper sites. Consequently, the overall conclusion that "our results suggest that AT/RT-specific methylation is linked to malfunctional SWI/SNF and PRC2 complexes" raises confusion.

In Figure 4F, again, the conclusion focuses on AR/RT hyper DMRs linking to reduced gene expression, which becomes complicated with PLEX samples. The PLEX samples also showed reduced target gene expression despite having hypo DMRs at the same linked regions. Only two genes (CXXC5 and TCEA3, as shown in Figure 4F) exhibited hyper DMRs linked gene decreases, specifically in AT/RT. Regarding NEUROG1 and NEUROD2, the authors also observed a decrease in expression in PLEX compared to MB. If DNA methylation is not involved (Figure 4G), what is the reason?

Part of the confusion arises from a lack of detailed explanation. In Figure 2C, processing the existing metadata and generating the depicted results need more clarity. Presumably, the authors utilized ChIP-seq data of individual TFs and chromatin regulators from different cell types (11 categories?) and overlaid this information with DMR sites from AT/RT and MB cancers. However, it is unclear how many DMR sites were used as input and what exactly is meant by "All" "Carcinoma" and multiple "Other" categories. Does "All" refer to all DMR sites, all TFs and chromatin regulator binding sites, or all cell type data utilized? Additionally, the categorization of "Other" cell types requires clarification.

The statement "As the SWI/SNF chromatin remodeling complex is incomplete in AT/RTs..." needs further explanation. Does this mean that the genome occupancies of the SWI/SNF chromatin remodeling complex in AT/RTs have yet to be studied?

The assessment of differentially expressed genes in AT/RTs, MB, and PLEX shown in Figure 4 needs to be clarified. Did the differentially expressed genes come from a comparison without control samples?

Kirsi Rautajoki (priorly Granberg)
Faculty of Medicine and Health Technology
Tampere University
Arvo Ylpön Katu 34
33520 Tampere
Finland
email: kirsi.rautajoki@tuni.fi
Tel: +358-50-3185819

To:
Editorial board
Life Science Alliance

Enclosed is our manuscript entitled “Aberrant DNA methylation distorts developmental trajectories in atypical teratoid/rhabdoid tumors” for your consideration for publication in Life Science Alliance. Based on the valuable comments of the reviewers, we have carefully revised the manuscript. This revision has included performing CUT&RUN experiment for AT/RT and medulloblastoma cell lines with key neural differentiation transcription factors as well as editing the manuscript based on the reviewers comments. More specifically, we analyzed NEUROD1 and NEUROG2 binding in AT/RTs and medulloblastomas using CUT&RUN method.

In the previous manuscript version, we showed how AT/RT-specific DNA hypermethylation is masking neural cell differentiation -related TF binding sites but, as the reviewers pointed out, there were not many measurements of TF binding in AT/RT or medulloblastomas available. We now showed that supportive results are obtained when NEUROD1 binding is measured in AT/RT and medulloblastoma cells. The NEUROG/NEUROD interplay, where some members of these families have altered expression and some have altered DNA methylation in their binding sites, is highly interesting. This is especially because these TFs cannot bind to methylated DNA although they include pioneer factors, which can open closed chromatin regions for gene activation. These results deepen our understanding of neural differentiation factor functions in AT/RTs.

In the manuscript, we have reported the genomic regions in which DNA methylation remains similar as in pluripotent cells or is unique to AT/RT. The detected pluripotent stem cell -like patterns are demethylated during differentiation, likely representing incomplete neural development -related demethylation. Our CUT&RUN experiment also provided verification for these results as there was no NEUROD1 binding in AT/RT in these genomic regions with pluripotent stem cell -like DNA methylation patterns. Furthermore, we saw that NEUROD1 was binding to these sites in medulloblastomas where the

sites are also hypomethylated. Differences in protein levels cannot explain the observed differences in NEUROD1 binding as it was similarly expressed in the both medulloblastoma and AT/RT cell lines.

We believe that our results will help the scientific community to better understand the potential causes and consequences of distorted cell differentiation in AT/RT tumorigenesis, which is likely to be a key driver of their malignancy.

We have responded to all the reviewers' criticism and comments below.

Reviewer #1 (Comments to the Authors (Required)):

Pekkarinen et al have submitted a manuscript on the role of DNA methylation in the control of gene expression in atypical teratoid rhabdoid tumor (ATRT), a lethal disease of the CNS that occurs in very young children. The mechanisms controlling this disease are reasonably well understood at this point, but there is certainly a need to better define epigenetic mechanisms controlling tumorigenesis. The manuscript is written reasonably well, though the numerous comparisons made in the paper are difficult to follow at times, and the manuscript is quite lengthy. I do not think language-specific editing is required. I have no concerns regarding conflicts of interest. Given the extensive reliance on normalization across studies, I would recommend the methods be thoroughly reviewed by a bioinformatician if the paper is considered for publication.

We thank the reviewer for the insightful comments and positive feedback. We have done our best to clarify the comparisons and the results in general. Furthermore, bioinformaticians have gone through the Methods section during revision. We want to point out that the sequencing and microarray datasets have been normalized and analyzed separately, and they have been integrated mainly through intersection. We have wanted to utilize this conservative approach to select robust features. We noticed that this has not been clearly described in the gene expression analysis section for array datasets in the methods, so this information has been added now (page 19, line 3).

In summary, the authors present an analysis of largely previously published DNA methylation and gene expression (microarray and RNA-Seq) in a set of ATRTs as well as medulloblastomas and choroid plexus tumors (pathology not defined) in order to define an ATRT-specific methylation signature. The authors use a bioinformatics analyses that leverage publicly available tissue and cell-line ChIP-Seq data to infer the impact of this methylation signature on transcription factor function, concluding that differentially methylated regions in ATRT block the activity of transcription factors and chromatin remodelers, which itself locks ATRT precursor cells into a more 'primitive' cell state.

Overall, the submitted manuscript contributes little new data to the literature (10 samples that underwent bisulfite sequencing and RNA sequencing). While the authors' concept of using DNA

methylation signatures to predict how SMARCB1 loss impacts lineage-specific transcription factor function is interesting, the analyses herein rely largely on circumstantial evidence. No experiments were performed to validate predictions made by bioinformatics analyses. More concerningly, there is no clear rationale for the comparisons made: there is no biological reason to use medulloblastoma or any choroid plexus tumor as a control tissue for ATRT. In this setting, I view the study as fundamentally flawed, but some of these could be addressed with substantial changes to the manuscript.

We thank you for this comment as it is important for the reader to understand the rationale. We were interested in finding which DNA methylation changes are specific to AT/RTs, as it harbors minimal genetic changes. We decided to compare AT/RTs with medulloblastomas and choroid plexus tumors, as they are also found in young children, thus developing early in life and being associated with embryonic development. Furthermore, they share common history, as most AT/RTs have been diagnosed as medulloblastomas before AT/RT became a tumor entity of its own, and there are also reports about INI1 negative choroid plexus tumors. Finally these tumors represent different developmental lineages, thus providing meaningful comparison points to each other.

From an experimental standpoint, I think the following would be necessary to sufficiently substantiate the authors' claims. There is a prevailing assumption throughout the manuscript that DNA methylation at transcription factor binding sites precludes their function in ATRT, and that differentially methylated regions control distinct transcription factor functions in ATRT and medulloblastoma, but there are no experimental data to support this conclusion in the manuscript. The authors focus on NEUROG1 and NEUROD2 as potential drivers of neural differentiation that are unable to function in ATRT due to their target sites being methylated, but no experimental data are presented to support this. I would recommend at least performing ChIP-Seq (or CUT&RUN) to these transcription factors and important enhancer-associated epigenetic marks (H3K27ac, H3K4me1, H3K4me3 and H3K27me3) as well as DNA methylation sequencing in either an experimental model of ATRT or in ATRT specimens and a control tissue (ie, normal brain or fetal brain, if available). I think the former approach would be both more feasible and mechanistically more convincing.

We thank the reviewer for this valuable comment, based on which we decided to perform CUT&RUN and reduced representation bisulfite sequencing (RRBS) experiments for AT/RT and medulloblastoma cell lines. In the manuscript, we report that several transcription factors harboring neural differentiation functions, such as NEUROD1 and NEUROG2, are enriched in areas that harbor AT/RT specific DNA hypermethylation. We decided to perform CUT&RUN experiment targeting NEUROD1 and NEUROG2 in 3 different AT/RT cell lines (CRL-3020, CRL-3036, CRL-3038) and 2 medulloblastoma cell lines (CRL-3021, Daoy).

As one of the medulloblastoma cell lines (Daoy) yielded no output DNA, we had to leave it out from the library preparation and, thus, analysis. Furthermore, Western plots with NEUROG2 antibody did not show a proper band and included clear background staining, which questions the specificity and affinity of the antibody. We decided to anyway perform CUT&RUN experiments with the NEUROG2 antibody, just in case, but only a limited number of peaks (maximum of seven after filtering) was detected per sample and none of them overlapped with our DMRs in

any of the cell lines. At RNA level, *NEUROG2* is less expressed in medulloblastoma than AT/RT, but we detected quite similar *NEUROG2* protein levels in the medulloblastoma cell line 3021 and the AT/RT cell lines when using another, CUT&RUN-incompatible antibody, so *NEUROG2* should be expressed in all the studied samples. Based on all this, we concluded that we are not able to provide any conclusive results and we most likely have issues with the *NEUROG2* CUT&RUN antibody, so we decided to leave this part out of the manuscript.

NEUROD1 experiments, however, gave us interesting results validating some of our previous findings. Some *NEUROD1* binding was detected in AT/RT cell lines but it did not overlap with differentially methylated regions. Importantly, *NEUROD1* binding sites in the medulloblastoma cell line partly overlapped with the regions harboring DNA hypermethylation in AT/RTs and hypomethylation in medulloblastomas in our tumor sample analyses. Additionally, when we studied DNA methylation in these same cell lines, we saw that *NEUROD1* binding sites from medulloblastoma cell line harbored clearly higher methylation in all three AT/RT cell lines compared to medullo cell line. Taken together, these findings support our hypothesis that DNA methylation in AT/RTs is blocking the DNA binding of *NEUROD1* whereas *NEUROD1* is recruited to certain genomic sites with lower DNA in medulloblastoma.

Interestingly, *NEUROD1* binding sites in the medulloblastoma sample overlapped especially with the regions whose DNA methylation in AT/RT resemble that in pluripotent stem cells and is demethylated during neural differentiation (from pluripotent stem cells to fetal brain). Thus, *NEUROD1* binding sites that were detected solely in medulloblastoma represent genomic regions that are linked to neural cell differentiation which appears to be blocked in AT/RT

Finally, we checked whether *NEUROD1* binding sites are detected in differentially methylated regions that are linked to differential gene expression (our DM-DE analysis). Notably, *NEUROD1* binding was measured in medulloblastoma from the *EBF3* gene locus which is hypomethylated in medulloblastomas when compared to AT/RTs with a concomitant upregulation of *EBF3*. Interestingly this gene plays a role in neural cell differentiation but it can also act as tumor suppressor. Furthermore it is known to be regulated by *NEUROD1* and the binding site we detected could mediate this regulation.

We also noticed, unfortunately quite recently, that GTRD contains *NEUROD1* binding sites measures from medulloblastoma cells and that we have had those binding sites in our analysis but, due to a human error, failed to categorize these as medulloblastoma origin in our last version of the categorized GTRD analysis. We now double-checked enrichments for this category and noted that *NEUROD1* measured from medulloblastoma cells is enriched in the AT/RT hypermethylated regions but not to those hypermethylated in MB or PLEX. This is in line with our previous enrichment results. As a note, similar results were obtained also for *OTX2*. The figures have been updated accordingly.

We think that the CUT&RUN experiments suggested by the reviewer have provided valuable new information that support our results and hypotheses in an earlier version of the manuscript and want to thank the reviewer for that.

Other major edits:

- Please provide a sound rationale for using medulloblastoma and choroid plexus tumors as control groups in the Results or Methods section. At the moment, this appears to be a comparison of convenience.

We thank the reviewer for this comment and understand that our approach might sound atypical. It is also important for the reader to understand the rationale. One reason has been the fact that we wanted to contrast tumor types instead of tumor subtypes, and one needs at least two references for detecting tumor-specific features. Medulloblastomas represent more differentiated tumors and can provide a good reference for AT/RT, although they nowadays are divided into several entities. We have provided additional explanations in the introduction (page 3, line 40) where we tell that these tumors not only share common history (have been confused to each other) but they also represent different developmental lineages. Thus, they represent meaningful comparison points to each other. We also believe that unconventional approaches are also often novel and might reveal new aspects about these pediatric tumors. We also want to point out that, in addition to comparing tumor types to each other, we also utilized pluripotent stem cells and fetal brain samples as references.

- For GTRD use, what restrictions were placed on the tissues / cell lines used in the analysis? TF binding sites are lineage-specific, so this has implications for how we interpret the output from this computational tool. For instance, it does not make sense to consider a previously validated EZH2 binding site in a pancreatic cancer to be considered a 'lost' EZH2 binding site in ATRT.

We agree that it is important to take into account that in which sample the TF binding has been measured. This is why we have grouped the data from GTRD based on the tissue type / cell line used. Furthermore, our CUT&RUN experiment also allowed us to validate results in the correct cell types. To clarify the categorized GTRD data analysis, we have now improved this part in the manuscript.

- The use of a manual literature search to annotate enriched TFs and chromatin remodelers into different 'themes' is vague and subject to selection bias. The authors provide no rationale as to why the selected proteins were chosen aside from their known roles in cell differentiation.

Thank you for this comment. We now mention in the manuscript more clearly how this annotation was made. We first identified the TFs and other transcriptional regulators that are enriched *e.g.* in AT/RT DNA hypermethylated regions and after that we annotate their functions, so selection of these TFs is fully data-driven. However we agree that many of the factors have several functions, and we might have not covered them all. As explained in the manuscript, TF theme analysis involved two annotation rounds: first, going TFs through individually and then again checking in a consistent manner whether an enriched TF can be linked to any of the themes once the themes were selected.

In manuscript it is now written (page 5, line 35) “To summarize the regulatory roles of the enriched TFs, we used a literature search to functionally annotate the TFs into different themes (Supplementary Table 2), which were chosen based on TF functions and gene families that

recurrently popped up when manually going through the enriched TFs in the first phase of the analysis (Merino et al, 2015; Swartling et al, 2015; Northcott et al, 2017; Erkek et al, 2019).”

- There are numerous instances throughout the manuscript where it is not clear whether the authors are referencing DNA methylation or H3K27 trimethylation. The authors should make all references to methylation clear, especially when discussing polycomb complexes.

We thank you for this comment as it is crucial to know which methylation is discussed. We went through each word “methylation” in the manuscript and made sure that it is clear whether we talk about DNA, H3K27, or some other histone, methylation.

- It is often difficult to follow the authors' references to which data they are analyzing. The comparisons made between ATRT and other tumor types / tissues seem arbitrary, and descriptions of findings often do not follow a specific line of inquiry. For instance, page 10, line 22: What is meant by the "split GTRD-data"?

We thank you also for this comment as it is important to know which data is discussed and how it has been produced. We have now done our best to clarify this throughout the manuscript. For example, “split GTRD data” refers to the analysis in which GTRD data has been categorized based on the sample type in which the DNA binding has been measured. So, in each category, DNA binding sites of TFs detected only in the samples representing that category (e.g. rhabdoid tumors) were included in the enrichment analysis. We acknowledge that the term “split GTRD data” has been confusing. Thus, we have now described the analysis better and renamed the outcome as “categorized GTRD data”.

Minor edits:

- The studies by Torchia and colleagues outlining the molecular subgroups of ATRT should be referenced in the manuscript, as these were seminal studies in defining our molecular understanding of ATRT.

This is indeed an important reference which had been dropped by accident. It has been now cited in the manuscript.

- Figure 1

o Why were the 10 matched RRBS / RNA-Seq samples included in the analysis? These are considered outliers in the tSNE plot?

We included these samples to our manuscript as they provided matched samples with both DNA methylation and gene expression data from the same tumor. This is valuable especially when considering connections between DNA methylation and gene expression. We do not consider these as outliers in the tSNE plot. The arrows used in the previous version of the figure (Figure 1B) were a little confusing as they are commonly used to point outliers, so we removed them from the figure. The point of the figure is to show that if we take genomic regions that have been measured both with RRBS and i450k, the RRBS samples are located next to the i450k samples representing the same tumor types. As these measurements differ and we are not using the exactly

the same CpGs for quantifying the DNA methylation, it is also expected that the RRBS samples tend to be located more on the borders of i450k clusters.

o Did the authors consider restricting their analysis to only ATRT vs. fetal brain and PSCs? And restricting the DMR analysis to ATRT subgroups only? This would be more likely to bring out subgroup-related differences.

We thank you for this question and yes, this was also considered. However we wanted to study especially features of the tumors that differed between tumor types. As PSCs are cultured cells, not tissue samples, we also get DMRs that are related to this difference in sample type, so we wanted to first compare tumor types to each other and then contrast only the detected DMRs to PSCs and fetal brain. We acknowledge that it is relevant to also study tumor subtypes separately, but that is already done by many and addresses different questions than comparison of tumor types.

- Figure 2:

o Only 'cancer-specific' DMRs were considered. How was this defined, and why weren't other DMRs considered to be of interest?

This was an important comment as it is crucial to know what “cancer-specific” DMRs means and why their selection is relevant. We defined cancer-specific DMR so that DMR has to be seen when compared to both of the two other tumor types. So *e.g.* AT/RT-specific hypermethylated DMR is hypermethylated in AT/RTs when separately compared to medulloblastomas and choroid plexus tumors. We did not include DMRs that were hypomethylated in one comparison and hypomethylated in the other, as we did not want to analyze regions whose DNA methylation is in between the two other tumor types. As can be seen from Figure 2A (right side of the four-field plots), such regions also presented a minority of all DMRs. We concentrated our analysis on these regions as their DNA methylation levels are specific for each tumor type and can, thus, clearly separate each tumor from both of the other tumors. This also streamlined the analysis, as if you run TF enrichments to each comparison separately, you are likely to have the same enriched TFs in several comparisons but you cannot be sure whether the enrichment is due to the same or different genomic regions. When using cancer-specific DMRs, we also detected the same enriched DMRs in several comparisons, but we can then be sure that there is no overlap between tumor-specific DMRs within hypermethylated or hypomethylated regions.

o The labeling of 'TFs' is misleading in Figure 3C; many of the labeled proteins / complexes are not TFs but rather chromatin remodeling complexes, histone deacetylases or their subunits (ie, ARID2, BRD9, SMARCA4, EZH2, EP300).

We thank you for this comment. It is true that not all the listed proteins / complexes are transcription factors and we have now done changes to clarify this. For simplicity, we are still using the term TF but define that it also covers other transcriptional regulators (page 5, line 18). We also use the term transcriptional regulators when needed in the text.

o P. 9, line 44, "Together these findings suggest that DNA methylation reduces the DNA binding of neural TFs, and thus neural differentiation, in AT/RT, 2 whereas a more advanced neural

differentiation is allowed in MB." This statement is not grounded in the data presented in the paper. No data regarding DNA binding of any of the listed proteins were presented in Figure 2.

Thank you for this comment. In Figure 2, we use DNA binding data of listed factors from the GTRD database, so this figure indeed includes DNA binding data, although not necessarily exactly from the studied tumor types. This has now partly improved as we also added the CUT&RUN data to this figure. We also acknowledge that the statement has been too strong, so it has been rephrased into: (page 7, line 4) "Together these findings suggest that the binding sites of neural TFs are hypermethylated in AT/RT, which might interfere with their DNA binding, whereas TF binding sites in PSCs are hypermethylated in MB, thus representing a more advanced neural differentiation state."

- Figure 3:

o What is the rationale for considering large-scale DMRs separately, and how are these defined?

Large-scale DMR definition is explained in more detail in methods (page 20, line 6). We took the topologically associating domains in neural progenitor cells and brain cortex and checked if there are at least five differential methylation (as defined for i450k and RRBS data) associated probes/tiles with a clear DNA methylation difference (0.25 in beta value scale) to the same direction in the tumor comparison in both the i450k and RRBS analysis. Simultaneously, less than 1/10 of differential methylation associated probes/tiles were allowed to undergo DNA methylation changes into the opposite direction within the same TAD. Moreover, the distance between the first and last probe/tile in the TAD had to be over 50 kb. DNA methylation is often expanded to the genomic surrounding, and TAD boundaries serve as boundaries, or blocks, for this expansion. However, DNA methylation can also target only specific, narrow sites or individual CpGs in the genome. We wanted to perform this analysis as large-scale DNA methylation has different regulatory outcomes compared to focal DNA methylation. Also, their upstream regulation, when considering what it takes to modify individual CpGs vs. several CpGs in a wider genomic region (e.g. during the extension of DNA hypermethylation), also differs from one another. We also made changes to the manuscript to make it clearer for the reader (page 5, line 7).

o "In conclusion, our results suggest that AT/RT-specific methylation is linked to malfunctioning SWI/SNF and PRC2 complexes" - This conclusion is not supported by the data presented in the manuscript, nor is this necessarily a novel concept. The authors frequently discuss 'methylation' in relation to PRC2 function in this section - are they referring to DNA or histone methylation, as polycomb complexes are generally thought to promote H3K27 trimethylation and are not known to have a DNA methylation function. Please clarify.

We thank you for this comment. This is related to an earlier comment where it was stated that it is not clear that which type of methylation is discussed in the manuscript. This part was one of the places where we made it clear which methylation we are referring to. We also agree that words used in this original conclusion did not match the data perfectly. We have changed this conclusion in our manuscript and it now states (page 10, line 31): "In conclusion, our results suggest that AT/RT-specific DNA methylation is covering the sites that are bound by SWI/SNF and PRC2 complexes in neural progenitors and more differentiated neural cells and that the DNA

hypermethylation in most of these sites is AT/RT-unique, not PSC-like. This aberrant DNA methylation has the potential to drive oncogenic epigenetic regulation, which is unique to AT/RT. In addition, we detected AT/RT-hyper regions that were similarly methylated in PSCs, harbor binding sites for several neural regulators, and were originally methylated via a PRC2-independent mechanism." We believe that this now better matches our results and provides insight for the reader.

- Figure 4

o The authors indicate methylation at predicted enhancers results in altered expression of target genes, but no experimental data are shown in this regard. Are there data to support the notion that DNA methylation alters enhancer activity? Or are the authors confusing DNA methylation with H3K27 trimethylation? The authors themselves acknowledge there is a mechanistic gap in this logic (page 18, line 27: "The balance between EZH2-induced H3K27me3 and DNA methylation is still unclear") ("Since 9 H3K27me3 levels and other studied histone marks are mainly depleted in AT/RTs (Erkek et al. 2019), the elevated DNA methylation is likely to play a role in AT/RT malignancy.")

We thank you for this comment. We are talking about DNA methylation in Figure 4. We systematically connected DMRs that are within the genomic neighborhood of a gene in the same topologically associating domain to the altered expression of that gene. In general, DNA methylation is known to alter transcription factor binding, most commonly by reducing the binding but sometimes also by increasing the binding of specific transcription factors. This has been shown to be the case also in the enhancer regions (*e.g.* Kreibich, Elisa, et al. "Single-molecule footprinting identifies context-dependent regulation of enhancers by DNA methylation." *Molecular Cell* 83.5 (2023): 787-802.). Active enhancers are also known to have reduced DNA methylation which supports the notion that DNA methylation alters enhancer activity. As some of these sites are listed also as **enhancers** that are **linked to certain genes** in FANTOM5 and/or GeneHancer database, we analyzed them separately, as well. In the quote, we are rather discussing whether EZH2, which is also reported to direct DNA methylation, has contributed to AT/RT unique DNA methylation instead of promoting H3K27 trimethylation and that it is unclear how these two non-overlapping EZH2 functions interplay with each other and how the balance between these different modes of action is determined.

Reviewer #2 (Comments to the Authors (Required)):

This study aimed to investigate the distinctions between atypical teratoid/rhabdoid tumors (AT/RTs), medulloblastomas (MBs), and choroid plexus tumors (PLEXs), which are brain tumors found in infants. These tumors are characterized by a low level of differentiation, leading to various cancer-related characteristics. The authors reanalyzed existing DNA methylation and gene expression datasets of AT/RTs, MBs, and PLEXs, along with samples from pluripotent stem cells (PSCs) and fetal brains. They examined DNA methylation patterns, associated DMRs with transcription factors (TFs) occupancy, and correlated with changes in gene expression to gain insights into the dissimilarities among these tumor types. In addition, they collected RRBS and RNA-seq data from 10 patients and integrated them into the analyses. Throughout the manuscript, the authors primarily focused on the results observed in hypermethylated differentially

methyated regions (DMRs) in AT/RTs. Cross-comparison between AT/RTs, MBs, and PLEXs would interest the brain cancer field as well.

Overall, the bioinformatic analyses in the manuscript are well-executed and the figures are presented effectively. However, some concerns need to be addressed before accepting the manuscript. These main points are outlined below:

Thank you for the positive comments concerning our approach, analyses, and figures.

1. Lack of full description of analyses and results in the main text: The authors should provide more information about their analyses and results in the main text. This would help readers better understand the methodologies employed and the outcomes obtained. Additionally, the authors should provide a clearer rationale and implications in the results section. For example, they could elaborate on the cell type classification and its implications for the analyses conducted using the Gene Transcription Regulation Database (GTRD). Similarly, the authors should explain the rationale and implications behind conducting analyses related to PSC/FB and differentiation. Some of the rationale and impact are currently discussed in the manuscript's discussion section - it would be more appropriate to move some into the results section.

We thank you for this comment as it is important for the reader to understand the rationale and analyses performed. We have improved our manuscript by better describing the rationale in the results section. We have explained in more detail how the GTRD data is used and how the cell type classification is taken into account (page 6, line 21) (“To take into account the stage of neural development, brain tumor-derived data, and cancer-association, we categorized GTRD data based on the sample type in which the binding data of the transcriptional regulators has been obtained.”). We also added information about the PSC/FB and differentiation and how/why they are analyzed (page 9, line 5) (“This allowed us to define whether tumor-specific DNA methylation levels actually reflect either low (PSC-like) or higher (FB-like) cell differentiation state or are unique to each tumor type, and whether they are altered during the differentiation from PSCs to FB”).

2. Lack of sufficient description of results related to MBs and PLEXs: Although the authors focused primarily on AT/RTs and hypermethylated differentially methylated regions (DMRs) in AT/RTs throughout the manuscript, there are also notable changes observed in medulloblastomas (MBs) (as evident in Figure 2B-C and Figure 3C-D). The authors need to provide more details about these results. It is essential to provide a comprehensive understanding of the findings.

Thank you for this comment. We have previously removed some of these results from our manuscript to be more focused and to save space. We have now added more detailed results also from the MBs and PLEXs where that is necessary. As MBs are more heterogeneous in their DNA methylation patterns, with clear differences between the tumor subtypes, we feel that they better serve as references for AT/RT analysis.

Specifically, in Figure 2 - to connect DMRs to biological functions, the authors focused on TFs motifs analyses in DMRs. Using CHIP-seq data depository and literature search, they sorted out TFs linked to neural differentiation and neural cell-related TFs uniquely enriched in AT/RT-hyper DMRs.

The authors overall concluded, "Together these findings suggest that DNA methylation reduces the DNA binding of neural TFs, and thus neural differentiation, in AT/RT, whereas a more advanced neural differentiation is allowed in MB." However, the analyses showed DNA methylation increases at pluripotency TF binding sites in MBs (Figure 2C), implying disrupted pluripotency in MBs. The authors should comment on the overall picture of what occurs in MBs.

Thank you for bringing this up. We added more MB results to this part of the manuscript. For example, we highlighted more the enrichment of the pluripotency TFs in MB DMRs and added what that could mean for these tumors.

Similarly, the statement "Out of these, 10 (67%) were enriched in AT/RT-hyper sites, MYCN in AT/RT-hypo, and only ZEB1 in MB-hyper regions". The authors did not mention the enrichment of REST and SIN3A in MB-hyper regions. Why?

This was due to the fact that only ZEB1 is both measured in 'other brain and neural cells' and significantly enriched in MB-hyper regions. The other neural TFs have been only measured or are only significant in other GTRD categories than those related to neural normal or tumor samples. We have now clarified this section and added a separate note about REST and SIN3A, whose binding sites in PSCs, but not those in 'other brain and neural samples' or 'other brain tumors', were enriched in MB hyper regions.

This kind of interpretation, which solely focuses on AT/RT hyper DMRs, continued throughout the study. In Figure 3D, the authors present DMR meta-analyses primarily centered around AT/RT hyper DMRs. However, a similar defect is observed in regions occupied by the SWI/SNF and PRC2 complexes in both AT/RT hyper and MB hyper sites. Consequently, the overall conclusion that "our results suggest that AT/RT-specific methylation is linked to malfunctional SWI/SNF and PRC2 complexes" raises confusion.

This is a good comment as there are similarities between AT/RT and MB results here. However, when considering the categorized GTRD analysis (Figure 2C), SWI/SNF and PRC2 members bind to different sites in GTRD categories and the TF enrichments in GTRD categories are largely different when the overlap is analyzed in respect to AT/RT-hyper vs MB-hyper DMRs. For example, the binding sites of SWI/SNF members ARID2, BRD9, and SMARCA4 in rhabdoid tumors are generally enriched in AT/RT-hypo and MB-hyper regions, whereas SMARCA4 binding sites in 'other brain and neural' samples are enriched in AT/RT-hyper and MB-hypo regions. When taking this all into account, the interpretation becomes more complicated, and presenting the MB results better would require quite extensive descriptions. We added a notion about SMARCA4 results to the manuscript (page 10, line 18) to follow the suggestions of the reviewer. In general, it is good to remember that there is no overlap between AT/RT-hyper and MB-hyper DMRs. This together with previously mentioned reasons to be more focused made us not to bring up the MB results more extensively in the manuscript.

In Figure 4F, again, the conclusion focuses on AR/RT hyper DMRs linking to reduced gene expression, which becomes complicated with PLEX samples. The PLEX samples also showed reduced target gene expression despite having hypo DMRs at the same linked regions. Only two genes (CXXC5 and TCEA3, as shown in Figure 4F) exhibited hyper DMRs linked gene

decreases, specifically in AT/RT. Regarding NEUROG1 and NEUROD2, the authors also observed a decrease in expression in PLEX compared to MB. If DNA methylation is not involved (Figure 4G), what is the reason?

We thank you for this comment, as discussion of the relationship of DNA methylation and gene expression is an important and interesting topic. DNA methylation can contribute to gene expression in different ways, for example by altering TF binding or changing chromatin structure. It can be stated that the transcriptional regulation in eukaryotes requires three things to happen: 1) chromatin should be accessible, 2) there should not be any other factors that hinder the interaction between TF and its DNA binding site, and 3) activating TFs should be present to drive the gene expression. DNA methylation can contribute to the regulation of the chromatin (1) and the binding site (2). But although DNA is ready for the interaction with TFs, it might be that the TFs (or at least some of them) are not expressed or they have been harvested somewhere else. If the activating regulator is missing, the gene is not properly expressed. Furthermore, it is possible that other suppressive mechanisms, such as H3K27me3, silence the gene activity in PLEXs although DNA methylation is decreased from the regulatory sites. Both of these explanations can contribute to the low expression levels detected in PLEXs, which we have also mentioned in the manuscript (page 12, line 4).

Part of the confusion arises from a lack of detailed explanation. In Figure 2C, processing the existing metadata and generating the depicted results need more clarity. Presumably, the authors utilized ChIP-seq data of individual TFs and chromatin regulators from different cell types (11 categories?) and overlaid this information with DMR sites from AT/RT and MB cancers. However, it is unclear how many DMR sites were used as input and what exactly is meant by "All" "Carcinoma" and multiple "Other" categories. Does "All" refer to all DMR sites, all TFs and chromatin regulator binding sites, or all cell type data utilized? Additionally, the categorization of "Other" cell types requires clarification.

We thank you for this comment as it is important for readers to understand what is discussed. We added clarifications of how data was used in this analysis and also changed some of our naming of the data to make it more clear what data is used where. Category 'All' has been now explained in the figure legend.

The statement "As the SWI/SNF chromatin remodeling complex is incomplete in AT/RTs..." needs further explanation. Does this mean that the genome occupancies of the SWI/SNF chromatin remodeling complex in AT/RTs have yet to be studied?

We were more referring to the fact that either SWI/SNF complex subunit SMARCB1 or SMARCA4 has been fully inactivated in AT/RTs, often via genomic deletion, which has been also clarified now in the manuscript: (page 7, line 11) "As SWI/SNF chromatin remodeling complex is incomplete in AT/RTs because of the full inactivation of SMARCB1 or SMARCA4..."

The assessment of differentially expressed genes in AT/RTs, MB, and PLEX shown in Figure 4 needs to be clarified. Did the differentially expressed genes come from a comparison without control samples?

Thank you for pointing out this unclarity. For the differentially expressed gene analysis, we did not have expression data from the normal brain controls, pluripotent stem cells or fetal brain, so only AT/RT, MB, and PLEX samples were used in the expression analysis. We have now clearly stated this in the manuscript (page 19, line 7).

February 28, 2024

RE: Life Science Alliance Manuscript #LSA-2023-02088-TR

Dr. Kirsi J. Rautajoki
Tampere University
Arvo Ylpön katu 34
Tampere 33520
Finland

Dear Dr. Rautajoki,

Thank you for submitting your revised manuscript entitled "Aberrant DNA methylation distorts developmental trajectories in atypical teratoid/rhabdoid tumors". We would be happy to publish your paper in Life Science Alliance pending final revisions necessary to meet our formatting guidelines.

- Please be sure that the authorship listing and order is correct
- Please upload all figure files as individual ones, including the supplementary figure files
- All figure legends should only appear in the main manuscript file, they should not be included with the figures. Please move figure legends at the end of manuscript file after the 'References'. Supplementary figure legends should be placed afterwards, as well as Supplemental Table legends. Please rename as: 'Figure legends', 'Supplementary figure legends' and 'Supplementary table legends'
- Please add a Category for your manuscript in our system
- Please add a Summary Blurb/Alternate Abstract in our system
- Please add the Twitter handle of your host institute/organization as well as your own or/and one of the authors in our system
- Please add ORCID ID for corresponding (and secondary corresponding) author--you should have received instructions on how to do so
- Please add an Author Contributions section to your main manuscript text
- Since there is only one single panel in Supplementary Figure 5, 11 and 15 please remove label A
- Please add a callout for the following Figures to your main manuscript text: S2A-J, S4A-B, S5B, S6A-B, S7A-C, S12A-B, S13A-B and S14A-D
- Accession number GSE197569 should be made publicly accessible at this time
- Please indicate whether written informed consent was obtained for the patient samples
- Joonas Uusi-Makela is indicated as the secondary corresponding author on the submission page but not in the manuscript, please update

A. FINAL FILES:

- An editable version of the final text (.DOC or .DOCX) is needed for copyediting (no PDFs).
- High-resolution figure, supplementary figure and video files uploaded as individual files: See our detailed guidelines for preparing your production-ready images, <https://www.life-science-alliance.org/authors>
- Summary blurb (enter in submission system): A short text summarizing in a single sentence the study (max. 200 characters)

including spaces). This text is used in conjunction with the titles of papers, hence should be informative and complementary to the title. It should describe the context and significance of the findings for a general readership; it should be written in the present tense and refer to the work in the third person. Author names should not be mentioned.

B. MANUSCRIPT ORGANIZATION AND FORMATTING:

Sincerely,

Reviewer #2 (Comments to the Authors (Required)):

The revised manuscript effectively addressed the concerns. The newly added NEUROD1 CUT&RUN data supports the functional outcome of the hypermethylated DMR in AT/RT and aligns with the impaired neuronal development observed in AT/RT. I believe the study is now suitable for publication in LSA.

March 6, 2024

RE: Life Science Alliance Manuscript #LSA-2023-02088-TRR

Dr. Kirsi J. Rautajoki
Tampere University
Arvo YlpÄ¶n katu 34
Tampere 33520
Finland

Dear Dr. Rautajoki,

Thank you for submitting your Research Article entitled "Aberrant DNA methylation distorts developmental trajectories in atypical teratoid/rhabdoid tumors". It is a pleasure to let you know that your manuscript is now accepted for publication in Life Science Alliance. Congratulations on this interesting work.

DISTRIBUTION OF MATERIALS:

Again, congratulations on a very nice paper. I hope you found the review process to be constructive and are pleased with how the manuscript was handled editorially. We look forward to future exciting submissions from your lab.

Sincerely,
